# Revealing the human mucinome

Stacy A. Malaker [1,4,5✉], Nicholas M. Riley [1,5], D. Judy Shon[1], Kayvon Pedram [1], Venkatesh Krishnan[2], Oliver Dorigo [2] & Carolyn R. Bertozzi [1,3✉]

Mucin domains are densely O-glycosylated modular protein domains found in various extracellular and transmembrane proteins. Mucin-domain glycoproteins play important roles in many human diseases, such as cancer and cystic fibrosis, but the scope of the mucinome remains poorly defined. Recently, we characterized a bacterial O-glycoprotease, StcE, and demonstrated that an inactive point mutant retains binding selectivity for mucin-domain glycoproteins. In this work, we leverage inactive StcE to selectively enrich and identify mucin-domain glycoproteins from complex samples like cell lysate and crude ovarian cancer patient ascites fluid. Our enrichment strategy is further aided by an algorithm to assign confidence to mucin-domain glycoprotein identifications. This mucinomics platform facilitates detection of hundreds of glycopeptides from mucin domains and highly overlapping populations of mucin-domain glycoproteins from ovarian cancer patients. Ultimately, we demonstrate our mucinomics approach can reveal key molecular signatures of cancer from in vitro and ex vivo sources.

[1] Department of Chemistry and Stanford ChEM-H, Stanford University, Stanford 94305 CA, USA. [2] Stanford Women's Cancer Center, Division of Gynecologic Oncology, Stanford University, Stanford 94305 CA, USA. [3] Howard Hughes Medical Institute, Stanford 94305 CA, USA. [4]Present address: Department of Chemistry, Yale University, New Haven 06511 CT, USA. [5]These authors contributed equally: Stacy A. Malaker, Nicholas M. Riley. ✉email: stacy.malaker@yale.edu; bertozzi@stanford.edu

Mucin domains are modular protein domains that adopt rigid and extended bottle-brush-like structures due to a high density of O-glycosylated serine and threonine residues[1–3]. Mucin-type O-glycans are characterized by an initiating α-N-acetylgalactosamine (α-GalNAc) monosaccharide that can be further elaborated into several core structures through complex regulation of glycosyltransferases[4,5]. As a result, mucin domains serve as highly heterogenous swaths of glycosylation that exert both biophysical and biochemical influence. For instance, this includes the ability to redistribute receptor molecules at the glycocalyx and to drive high avidity binding interactions[6–8]. In the canonical mucin (MUC) family, mucin domains often occur as tandem repeats, creating heavily glycosylated superstructures. Canonical mucins are central to many functions in health and disease, and have long been associated with human cancers, e.g., MUC1 and MUC16 (also known as CA-125)[9–12]. Dysregulation of mucin domain expression and aberrant mucin domain glycosylation patterns have been implicated in disease pathologies, especially in tumor progression, where mucins modulate immune responses and also promote proliferation through biomechanical mechanisms[13–15].

Mucin domains also exist in proteins outside of the 21 canonical mucins (Fig. 1A). For example, CD43 on the surface of leukemia cells selectively interacts with the glyco-immune checkpoint receptor Siglec-7 through its N-terminal mucin domain[16]; mucin domain-containing splice variants of CD44 (CD44v) serve as cancer cell markers relative to the ubiquitously expressed standard isoform[17]; CD45 mucin domains act as suppressors of T-cell activation[18]; mucin domain O-glycosylation on PSGL-1 is required for leukocyte-endothelial interactions[19]; and aberrant regulation of mucin domains in podocalyxin and SynCAM1 are implicated in a variety of cancers[20,21]. In all of these cases, shared functional attributes of mucin domains impart structural and biophysical properties relevant to their biology. Thus, instead of the more traditional categorization of the glycoproteome into N- and O-glycoproteins (both of which are represented by mucin-domain glycoproteins), it is logical to parse the glycoproteome into the mucinome, a family of glycoproteins whose mucin domains make them functionally related. However, even as the tools to capture the broadly defined N- and O-glycoproteome continue to improve[22–31], mucin domains remain enigmatic and difficult to characterize. As such, a comprehensive list of all proteins with a mucin domain does not exist. This lack of a well-defined mucinome leaves a critical blind spot in our ability to interrogate mucin domain functions across molecular biology.

Toward this goal, enzymes derived from microorganisms known to colonize mucosal environments have shown promise for developing tools specifically suited to characterize mucin-domain glycoproteins[32–38]. We recently characterized a panel of such enzymes, termed O-glycoproteases, and showed that each of them harbor a selectivity toward mucins as well as unique peptide- and glycan-based cleavage motifs[39]. Using catalytic point mutants, we also demonstrated that select O-glycoproteases can retain binding specificity for mucin domains; these were then used as mucin-selective staining reagents for Western blots, immunohistochemistry, and flow cytometry[39]. One particular enzyme of interest is secreted protease of C1 esterase inhibitor (StcE) from enterohemorrhagic *Escherichia coli*, which recognizes mucin domains decorated with a variety of O-glycan modifications[40–43]. This gives StcE both the selectivity needed to specifically bind mucin domains and the breadth to bind diverse mucin domain subtypes that vary in glycosylation patterns. Indeed, StcE has shown great utility for selective release of mucin fragments from biological samples and for improving mass spectrometry (MS)-based analysis of mucin domains[40].

We reasoned that the catalytically inactive point mutant of StcE (StcE[E447D]) could function as a universal mucin enrichment tool for mucin domain discovery, similar to how inactive O-glycosidases and engineered sialidases can enrich broadly for O-glycosylated and sialylated glycoproteins, respectively[44,45]. Here we show that StcE[E447D]-conjugated beads selectively enrich mucin-domain glycoproteins from complex cancer cell lysates and from crude ovarian cancer patient ascites fluid. As part of this workflow, we developed a mucin-domain candidacy algorithm to assign confidence scores to proteins that have a high likelihood of containing a mucin domain. Additionally, we detected hundreds of glycopeptides derived from mucin domains in the StcE[E447D]-enriched samples. Ultimately, we demonstrate that this mucinomics platform can define key molecular signatures of cancer in both in vitro and ex vivo systems and is a valuable approach to unravel the role of mucin domains in health and disease.

## Results

**Mucin enrichment and definition strategy to describe the mucinome.** Our previous work indicated that a catalytically inactive point mutant of StcE (StcE[E447D]) retains its binding specificity for mucin domains while leaving them intact for subsequent analysis[39,40]. Through a straightforward reductive amination approach, we conjugated StcE[E447D] to POROS-AL beads to generate a solid phase support material to use for enrichments[46]. To optimize our enrichment protocol, we added StcE[E447D]-conjugated beads to OVCAR3 supernatant followed by an anti-MUC16 Western blot for detection. We tuned several parameters of the enrichment, including binding time, bead-to-substrate ratio,

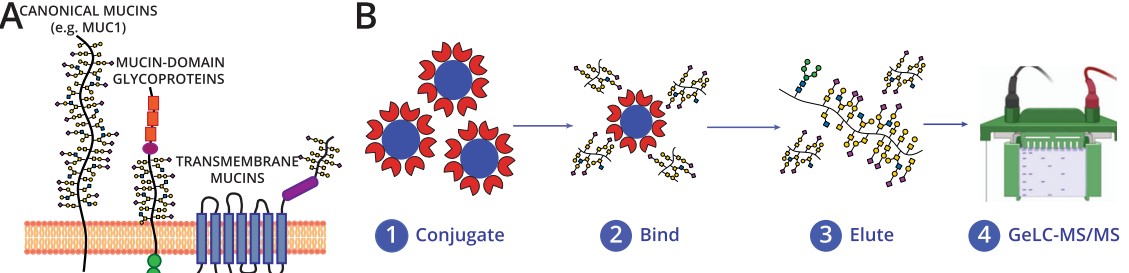

**Fig. 1 Mucinomics platform for enrichment of mucin-domain glycoproteins in complex samples. A** The mucinome comprises a variety of proteins that have a densely glycosylated mucin domain. Mucin domains are found in canonical mucins, mucin-domain glycoproteins, and even multi-pass transmembrane proteins. **B** Workflow for enrichment technique. StcE[E447D] beads were conjugated using reductive amination to POROS-AL 20 beads, followed by capping in Tris-HCl (1). Complex samples (lysate, ascites) were added to the beads and allowed to bind overnight (2), washed, and eluted by boiling in protein loading buffer (3). Samples were fractionated via one-dimensional gel electrophoresis and digested in-gel using trypsin (4); the gel electrophoresis chamber was created with BioRender.com.

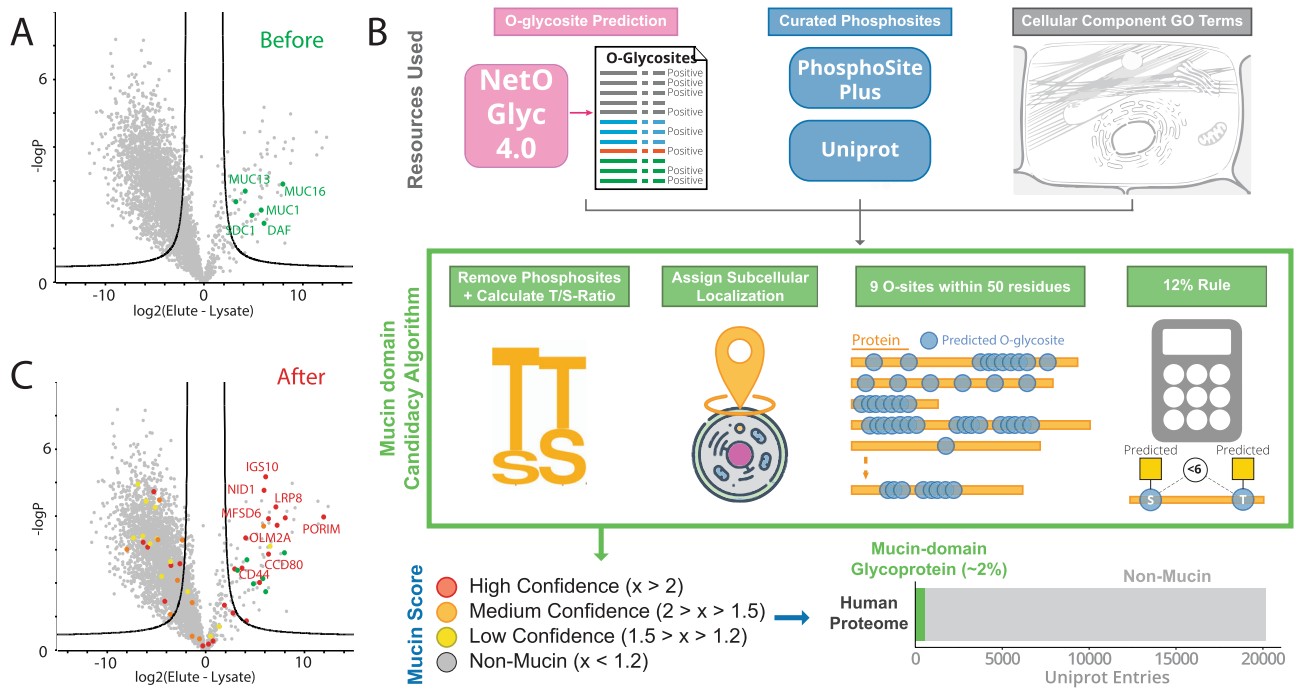

**Fig. 2 Mucin-domain candidacy algorithm for confident assignment of mucin-domain glycoproteins. A** Known mucins in HeLa lysate enrichment. HeLa lysate was subjected to the enrichment procedure described in Fig. 1 and known mucin-domain glycoproteins (MUC1, MUC13, MUC16, DAF, and SDC1) were labeled. Source data are in Supplementary Data 3. Significance testing was performed using a two-tailed t-test with 250 randomizations to correct for multiple comparisons, an FDR of 0.01, and an S0 value of 2. **B** Mucin-domain glycoprotein candidate annotation. A mucin-domain candidacy algorithm was created to assign Mucin Scores to indicate confidence that a given protein contains a mucin domain. First, predicted O-GalNAc sites were generated by the NetOGlyc4.0 tool, curated lists of phosphosites were downloaded from PhosphoSitePlus and Uniprot, and cellular localization GO terms were downloaded. The mucin-domain candidacy algorithm then removed predicted O-GalNAc sites overlapping with known phosphosites, calculated the proportion of threonine to serine residues (T/S-ratio), evaluated protein subcellular localization, and checked for frequency and density of predicted O-GalNAc sites. These metrics were used to calculate a Mucin Score, which could then be used to evaluate mucinome enrichment. The entire human proteome was processed with the mucin-domain candidacy algorithm; using manually curated benchmarks, 357 proteins have mucin domains (~2% of human proteome). The cell image is licensed through a CC BY 4.0 license from the Uniprot database[52]. **C** Mucinome of HeLa lysate. The results in **A** were processed with the mucin domain definition program, and mucin-domain glycoproteins were labeled according to the Mucin Score. Red signified a score of >2 (high confidence), orange 2–1.5 (medium confidence), and yellow 1.5–1.2 (low confidence). Known mucin-domain glycoproteins labeled in **A** are still labeled in green. Source data are in Supplementary Data 3. Significance testing was performed using a two-tailed t-test with 250 randomizations to correct for multiple comparisons, an FDR of 0.01, and an S0 value of 2.

wash buffers, and elution conditions (see Methods); a simplified protocol is detailed in Fig. 1B. With a suitable enrichment protocol defined, we scaled up the reaction for mass spectrometry by enriching 500 µg of HeLa cell lysate with 100 µL of pre-washed StcE[E447D]-conjugated beads. Bound proteins were eluted by boiling in protein loading buffer, elutions were separated by one-dimensional gel electrophoresis, and in-gel digestions were performed prior to label-free quantitative shotgun proteomics (i.e, GeLC-MS/MS, see Supplementary Fig. 1).

To calculate the degree of enrichment provided by StcE[E447D]-conjugated beads, 30 µg of unenriched cell lysate was simultaneously prepared and analyzed alongside each elution. Significantly enriched proteins were determined by comparing area-under-the-curve-based label free quantitation (LFQ) values for proteins in the elution relative to lysate, with processing and calculations performed using MaxQuant and Perseus[47,48]. The volcano plot in Fig. 2A shows several known and canonical mucins enriched in the elution (right; green), as opposed to untreated lysate (left). In particular, MUC1, MUC13, and MUC16 were significantly enriched, as well as known mucin-domain glycoproteins CD55 (decay-accelerating factor, DAF) and syndecan-1 (SDC1).

While these initial results were exciting, it quickly became clear that hand-curating proteins with known mucin domains would

be untenable for the mucinome discovery platform. Not only is hand-curation low throughput, but it inherently misses proteins without known mucin domains. Instead, we developed a mucin-domain candidacy algorithm to calculate which proteins have a high probability of bearing a mucin domain. Previous work has mined sequences looking for PTS domains in various non-human organisms[49,50], but we wanted to extend our criteria to use protein-level data that includes predicted O-glycosites, subcellular localization information, and previously annotated PTM-sites to annotate putative mucin domains in the human proteome. As summarized graphically in Fig. 2B, our algorithm comprised several steps to assign a Mucin Score to every protein in the human proteome. Mucin-domain candidacy algorithm processing was preceded by O-GalNAc glycosite prediction using the NetOGlyc4.0 tool, a support vector machine-based predictor developed using a map of ~3,000 O-glycosites from 600 O-glycoproteins that was generated through SimpleCell technology[51]. Predictions from NetOGlyc4.0 were then screened for known phosphosites annotated in Uniprot[52] and PhosphoSitePlus[53], and any overlap in phosphosites with predicted O-GalNAc sites resulted in removal of the predicted O-GalNAc site from consideration. This was a necessary step because NetOGlyc4.0 often predicted O-GalNAc sites in known phosphodomains of intracellular proteins, resulting in a high

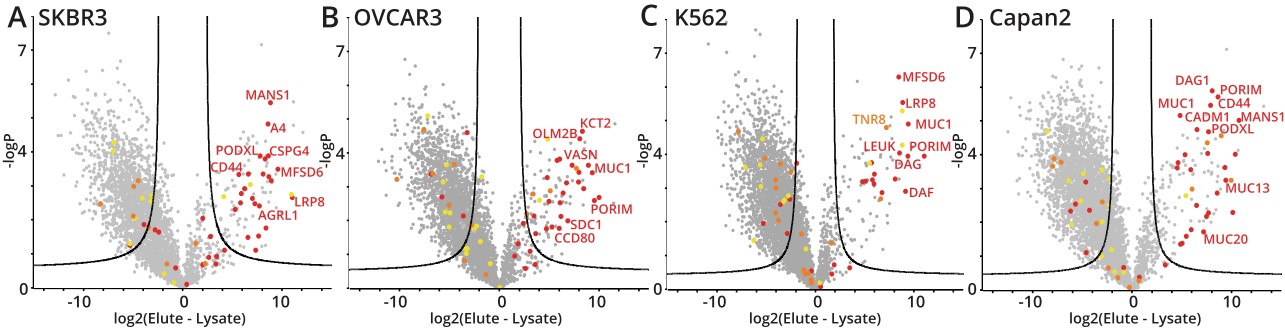

**Fig. 3 Mucinome of cancer-associated cell lines. A–D** Volcano plots representing four enrichment experiments from SKBR3 (**A**), OVCAR3 (**B**), K562 (**C**), and Capan2 (**D**) cell lines. Cell lysates were subjected to the workflow described in Figs. 1 and 2, scored with the mucin-domain candidacy algorithm, and mucin-domain glycoproteins were labeled according to the Mucin Score. Red signified a score of >2 (high confidence), orange 2–1.5 (medium confidence), and yellow 1.5–1.2 (low confidence). Strongly enriched proteins were labeled with their gene names associated with specific proteins. Source data are in Supplementary Data 3. Significance testing was performed using a two-tailed t-test with 250 randomizations to correct for multiple comparisons, an FDR of 0.01, and an S0 value of 2 for all volcano plots.

number of false positive mucin candidates after downstream processing. Note that O-GalNAc and phosphorylation sites are not known to have a high degree of overlap, as the former is generally extracellular whereas the latter is often intracellular.

Following O-GalNAc site prediction and phosphosite filtering, the algorithm asked four questions of each protein: (1) Was the protein predicted to be extracellular, secreted, and/or transmembrane?; (2) Were there at least 9 predicted O-glycosylation sites within a stretch of 50 residues?; (3) Was the distance between any given pair of O-glycosites less than 12% of the entire mucin domain (i.e., are glycosites <6 residues away from each other in a 50 residue sequence)?; and (4) Was the ratio of threonine to serine residues skewed toward threonine? Each of these benchmarks were determined through expert curation of known mucin sequences, which are further described in *Methods*. Using a point system based on the answers to these questions, the algorithm ultimately assigned a Mucin Score to each protein in the human proteome. By manually assessing outputs, we determined that a score of >2 was a high confidence mucin-domain glycoprotein, between 2 and 1.5 was a medium confidence mucin-domain glycoprotein, and between 1.5 and 1.2 was a low confidence mucin-domain glycoprotein. Proteins with a score lower than 1.2 were not considered mucin-domain glycoproteins. Levels of confidence also capture the idea that a mucin domain may not be a binary concept; there may be gradients of O-glycosylation density and patterns that contribute to mucin-like attributes. See Supplementary Data 1 for the mucin candidate algorithm output of the entire human proteome and Supplementary Data 2 for the location of where putative mucin domains and predicted O-glycosites occur; 357 proteins contain a putative mucin domain by our estimate (score > 1.2), encompassing 20 of the 21 canonical mucins (MUC-15 was excluded), and comprising roughly 2% of the proteome (Fig. 2B). For comparison, proteases represent up to 2% of the human proteome; thus, mucin-domain glycoproteins could be much more common than previously thought[54].

Using Mucin Scores to reannotate the dataset from Fig. 2A, we labeled high, medium, and low confidence mucin-domain glycoproteins as red, orange, and yellow, respectively (Fig. 2C). The canonical and known mucins from Fig. 2A are still labeled in green. A large number of high confidence mucin-domain glycoproteins are enriched in the StcE$^{E447D}$ elution, some of which are labeled in red with gene names associated with specific proteins. Interestingly, some high confidence and a handful of medium to low confidence mucin-domain glycoproteins are on the left side of the volcano plot, i.e., not enriched in

the StcE$^{E447D}$ elution. This could indicate (1) that StcE$^{E447D}$ does not effectively enrich some mucin domains, (2) that mucin domains in these proteins are not heavily glycosylated in HeLa cells, or (3) that the mucin-domain candidacy algorithm has some degree of error. Inherently, the mucin-domain candidacy algorithm is an imperfect predictor of all mucin domains across the proteome. Indeed, no high efficacy mucin domain prediction algorithm exists, nor was that the focus of this work. Instead, our mucin-domain candidacy algorithm indicates degrees of confidence for assigning proteins with a putative mucin domain that can be used to assess mucinome enrichment with StcE$^{E447D}$-conjugated beads.

**Inactive O-glycoproteases enrich mucin-domain glycoproteins from various cancer cell lines.** Given that the HeLa lysate enrichment was successful, we decided to expand the approach to other cancer-associated cell lines, including SKBR3 (breast), OVCAR3 (ovarian), K562 (leukemia), and Capan2 (colorectal). The corresponding volcano plots are shown in Fig. 3A–D (see Supplementary Data 3 for Perseus processing files). As before, red dots signified a score of >2 (high confidence), orange dots 2–1.5 (medium confidence), and yellow dots 1.5–1.2 (low confidence). Strongly enriched mucin-domain glycoproteins were labeled with their gene names associated with specific proteins.

The Upset plot in Fig. 4A compares commonly observed mucin-domain glycoproteins across the cell lines. The total number of enriched mucin-domain glycoproteins from each cell line is shown on the bottom left (blue horizontal bars). If a group of mucin-domain glycoproteins was only seen in one cell line, only one gray dot is darkened; the number of proteins that are only seen in that cell line are shown in bar graph form above. For instance, 9 mucin-domain glycoproteins were only detected in the K562 cell line. Overlap between samples are shown by multiple darkened gray dots and a line connecting them; as an example, 2 mucin-domain glycoproteins were only detected in both the SKBR3 and OVCAR3 cell lines. A total of seven mucin-domain glycoproteins were seen in all five cell lines; these proteins are shown above the Upset plot. The putative mucin domain (orange, as calculated by the mucin-domain candidacy algorithm), transmembrane domains (purple), and annotated N-glycan sites (green) are noted on each of the proteins.

To better understand how many of the proteins contained previously undescribed mucin domains, we compared our dataset to the SimpleCell dataset from Clausen and colleagues[51], which is one of the most comprehensive study on O-glycosites to date (albeit with truncated O-glycan species). To consider a

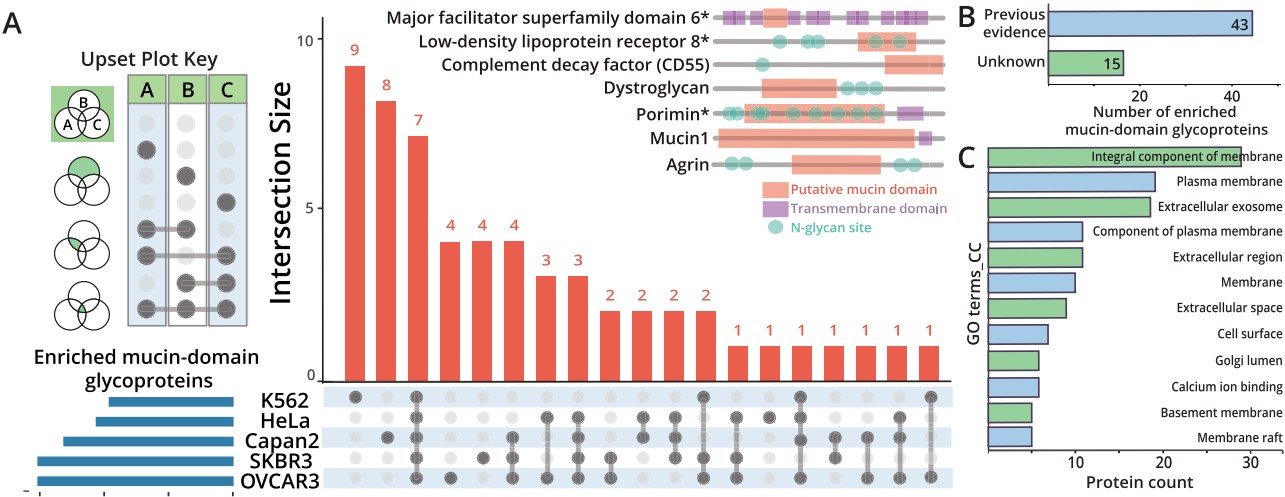

**Fig. 4 Analysis of mucin-domain glycoproteins from cell line enrichments. A** Upset plot comparing enriched mucin lists from five cell lines. The total number of enriched mucin-domain glycoproteins from each cell line is shown on the bottom left (blue horizontal bars). If a group of mucin-domain glycoproteins was only seen in one cell line, only one gray dot is darkened; the number of proteins that are only seen in that cell line are shown in bar graph form above. Overlap between samples are shown by multiple darkened gray dots and a line connecting them. A total of seven mucin-domain glycoproteins were seen in all five cell lines; these proteins are shown above the Upset plot. The seven proteins found in all five samples are shown above the plot; putative mucin domain (orange), transmembrane domains (purple), and N-glycan sites (green) are annotated based on Uniprot assignments (transmembrane domain, N-glycan sites) and the mucin-domain candidacy algorithm (mucin domain). **B** Discovery of new mucin glycoproteins. The total list of enriched mucin proteins was searched against the SimpleCell dataset[51]. If O-glycopeptides within the assigned mucin domains were found in this dataset, a protein was considered as a previously described mucin-domain glycoprotein; if not, it was considered a previously undescribed mucin-domain glycoprotein. **C** Mucin GO term enrichment. Enriched cellular component (CC) GO_terms using DAVID[55,56] are shown with protein count indicated on the x-axis. Source data are in Supplementary Data 4 and 5.

mucin-domain glycoprotein in this comparison, more than 1 glycopeptide had to be detected from within the assigned mucin domain. Additionally, if the protein was a canonical (e.g. MUC15) or confirmed (e.g. Gp1bα) mucin-domain glycoprotein, these were considered as previously described/known proteins. Several of the proteins (4/7) found in all five cell lines were previously known to have a mucin domain, including: Mucin-1 (MUC1), dystroglycan (DAG1), agrin (AGN), and complement decay factor (CD55, DAF). However, we discovered that three of the overlapping proteins have previously undescribed mucin domains: low-density lipoprotein receptor 8 (LRP8), major facilitator superfamily domain 6 (MSFD6), and porimin (PORIM). MSFD6 is a multi-pass transmembrane protein that is implicated in antigen processing and presentation of exogenous peptide antigens via MHC class II, whereas porimin is involved in oncotic cell death characterized by vacuolization and increased membrane permeability.

Extending this analysis to all of the enriched mucin-domain glycoproteins, we found that approximately one-quarter (~31%, 15 of 58) were newly discovered (Fig. 4B, Supplementary Data 4). Of these proteins, perhaps the most surprising was adhesion G protein-coupled receptor L1 (ADGRL1), as GPCRs are generally not thought of as mucin-domain glycoproteins. This particular GPCR is implicated in both cell adhesion and signal transduction; future studies will be devoted to understanding the role of mucin domains in GPCR signaling. To broadly characterize features and functions of proteins present in our mucinome list, we performed GO term enrichment using DAVID[55,56]. Perhaps unsurprisingly, the most enriched cellular component (CC) GO terms were associated with membranes, cell surfaces, extracellular space, among others (Fig. 4C).

In another extension of our mucinomics workflow, we performed an enrichment using a different O-glycoprotease. While StcE does not demonstrate drastic glycan specificity, we have characterized several other O-glycoproteases with varying glyco-proteolytic specificities[39]. BT4244 is a O-glycoprotease of particular interest from *Bacteroides thetaiotaomicron* that cleaves N-terminally to serine and threonine residues bearing truncated O-glycans, such as the cancer-associated T- and Tn-antigens (Gal-GalNAc and GalNAc, respectively). We reasoned that a point mutant of BT4244 (BT4244$^{E575A}$) could also enrich mucin-domain glycoproteins bearing shortened O-glycan structures. Thus, we conjugated BT4244$^{E575A}$ to beads and performed an analogous enrichment using HeLa lysate with and without sialidase pretreatment, with results shown in Supplementary Fig. 2. Without sialidase treatment, only six mucin-domain glycoproteins were significantly enriched in the elution, suggesting that not many mucin-domain glycoproteins bear truncated O-glycans in HeLa cells. We then pre-treated HeLa lysate with 100 nM sialidase overnight and repeated this procedure, which resulted in the enrichment of 13 mucin-domain glycoproteins. Though not as robust as StcE enrichment, this proof-of-principle procedure demonstrates that other O-glycoproteases could be used to enrich and identify cancer-associated glycoforms of mucin-domain glycoproteins.

We next asked how selective our mucin-domain-centric platform is when compared to lectin (i.e,. glycan-centric) enrichments commonly used for O-glycoproteomics. Jacalin has preference for mucin-type O-glycans including GalNAc and GalNAc-Gal; thus, we conjugated Jacalin to POROS-AL beads and performed enrichments on HeLa cell lysate with and without pretreatment with sialidase. The resulting volcano plots are shown in Supplementary Fig. 3. To be sure, Jacalin does enrich most of the mucin-domain glycoproteins, but as demonstrated by the large number of enriched non-mucin proteins, it is clear that Jacalin is less specific for mucin-domain glycoproteins. This point is further illustrated in Supplementary Fig. 4. The Jacalin (+/− sialidase) pulldown resulted in the enrichment of 205 and 273 proteins, respectively. The percentage of mucin-domain glyco-proteins within this subset is only 16–17%, meaning that 171 and

230 non-mucin proteins were enriched in the two samples. Using the same HeLa lysate, StcE[E447D]-conjugated beads enriched a total of 75 proteins, 28% of which were mucin-domain glycoproteins. Thus, StcE[E447D] is approximately two-fold more selective for mucin-domain glycoproteins. Further, we detected only 54 non-mucin proteins in this enrichment, compared to the 230 in the Jacalin pulldown, representing a > 4-fold reduction in non-mucin proteins. While Jacalin did enrich more mucin-domain glycoproteins, selectivity is especially important when considering potential goals of characterizing mucin-domain O-glycopeptides; non-mucin proteins, and their associated unmodified peptides, will outcompete the glycopeptides for ionization and detection.

We then investigated the non-mucins that were enriched by the StcE[E447D] cell line enrichments to understand if there was an unexpected selectivity for features other than mucin domains or if it was likely due to non-specific binding. We calculated how many of the non-mucins were commonly found between cell lines, as demonstrated by the Upset Plot in Supplementary Fig. 5. Here, the majority of enriched proteins were found in only one cell line, suggesting that these proteins were primarily non-specifically binding to the beads. On the other hand, 5 proteins were found in all cell lines, and 7 were found in at least 4 cell lines (Supplementary Data 5; Master_NonMucin tab). Of these 12 proteins, 6 are potential mucin-domain glycoproteins with Mucin Scores that did not meet our initial thresholds but have several predicted O-glycosylation sites. The other proteins are likely to be (a) abundantly expressed and non-specifically binding (e.g. myosin) and/or (b) previously undescribed glycan or mucin-binding proteins. Taking this one step further, we performed cellular component GO term enrichments for all of the non-mucins. The highest protein counts were "extracellular exosome" (87) and "integral component of membrane" (80); "perinuclear region of cytoplasm" was far less abundant at a protein count of 15 (Supplementary Data 5).

Additionally, we explored which assigned mucin-domain glycoproteins were repeatedly not enriched by our technique. As with the enriched non-mucins, we generated an Upset Plot to determine which of our assigned mucin-domain glycoproteins were not enriched reproducibly (Supplementary Fig. 6). Here, five proteins were consistently not enriched across all five cell lines and five across at least four cell lines. The majority of these proteins were intracellular cytoplasmic proteins that were likely overscored as mucin-domain glycoproteins due to their presumed phosphorylation/O-GlcNAc sites that were predicted by NetO-Glyc4.0 as O-GalNAc sites (Supplementary Data 6). We tried to account for these proteins by removing annotated phosphosites from the NetOGlyc4.0 glycosite assignments, though, we note that phosphosite databases are likely incomplete. Taken together, we believe that these analyses demonstrate that our approach provides satisfactory selectivity for mucin domains.

**Mucinomics platform allows for identification of ovarian cancer patient mucinome**. Following the establishment of our mucin domain enrichment approach in cell lines, we next wanted to test the mucinomics platform on clinically relevant patient samples. Ovarian cancer ranks fifth in cancer deaths among women and is often diagnosed in stage III or IV, leading to a poor prognosis. This is due, in part, to the fact that the only clinically relevant biomarker is CA-125, a peptide epitope of MUC16, but the exact structural definition of this antigen continues to be elusive. Previously, we showed that StcE could digest MUC16 from crude ovarian cancer patient ascites fluid[40], leading us to reason that our enrichment technique could be used to selectively isolate MUC16 and other mucin-domain glycoproteins from

ascites fluid as a potential diagnostic strategy. As such, we performed mucinomics enrichment with StcE[E447D]-beads on five de-identified patient samples (OC235, OC234, OC114, OC109, and OC107). As seen in Fig. 5A–E, the grand majority of putative mucin-domain glycoproteins were significantly enriched in the elution (see Supplementary Data 7 for Perseus processing information); in all but one of the experiments (OC114), MUC16 (denoted in purple) was significantly enriched. The enrichment in these experiments was even more successful than in the cell lines; in four out of five patient samples (excluding OC235), zero mucin-domain glycoproteins were "enriched" in the crude ascites fluid. This is also demonstrated by the selectivity calculations depicted in Supplementary Fig. 7, as well as the non-mucin proteins investigated in Supplementary Fig. 8 and Data 8. For the full list of enriched mucin-domain glycoproteins, see Supplementary Data 8. The enrichment was likely more successful due to the presence of fewer interfering proteins found in biofluids. Again, we compared our results to the SimpleCell dataset and found approximately half (~54%, 33 of 61) of the mucin-domain glycoprotein candidates have previously unannotated mucin domains; these are detailed in Supplementary Data 4.

Figure 5F compares overlap between the ascites samples with a Venn diagram of all enriched mucin proteins. Each sample is represented by a different color box, and the overlap between samples is given by a number within the boxes. Notably, 26 mucin-domain glycoproteins were enriched in all five samples, demonstrating substantial overlap between patients. The 26 overlapping proteins and their MucinScores are listed in Supplementary Table 1. Again, as expected, the most enriched cellular component GO terms for the mucin-domain glycoproteins were associated with membranes, lumen, extracellular matrix, and the basement membrane (Fig. 5G). As before, the mucinome list contains some known mucin-domain glycoproteins, such as CD44, podocalyxin (PODXL), and agrin (AGRN). In addition, the list contains previously undescribed mucin-domain glycoproteins, such as thymosin beta-4 and Trem-like transcript 2 protein. This further underscores the need for tools, like the strategy described here, to help define members of the mucinome. Additionally, we detected adhesion G protein-coupled receptor L1 (ADGRL1) as enriched in all five samples, further enforcing our conviction that this protein contains a mucin domain. While our patient cohort is currently too small to make any clinical claims, we believe that these overlapping mucin-domain glycoproteins could represent a better diagnostic and/or prognostic indicator for ovarian cancer. Future efforts will be devoted to expanding the study to a larger number of patients and comparing the results to patient outcomes, with the goal of developing a rapid mucin-fingerprinting approach using this mucinomics platform.

**StcE[E447D]-enrichment also captures O-glycopeptides from mucin domains**. Characterization of intact O-glycopeptides was not an original goal when designing these experiments, but we reasoned that StcE[E447D]-enrichment should function as a *de facto* glycopeptide enrichment by selecting for highly O-glycosylated mucin-domain glycoproteins at the protein (i.e, pre-proteolysis) level. We observed a large number of spectra in our ascites enrichments bearing the "HexNAc fingerprint", that is, oxonium ions specific to glycopeptides, which prompted us to search our data for intact glycopeptides. Generally, electron-driven dissociation is better suited for characterizing O-glycopeptides because it can provide O-glycosite localization[57,58]. This is especially true for O-glycopeptides derived from mucin-domain glycoproteins, which will likely have multiply glycosylated sequences[59–61]. Even so, collision-based fragmentation can still

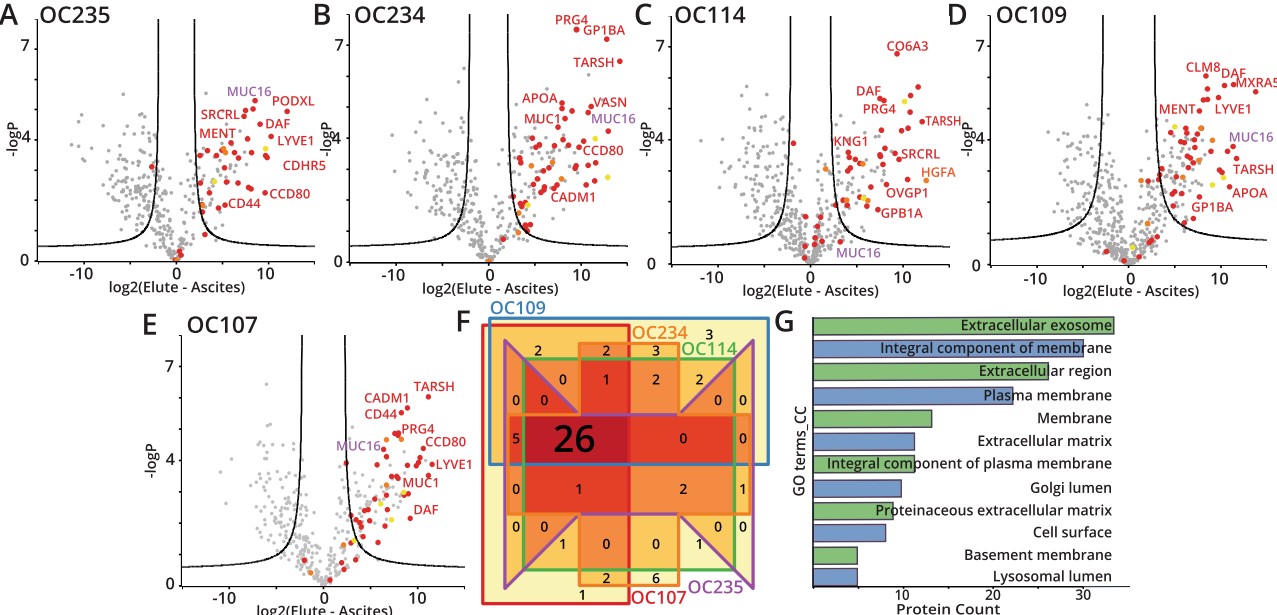

**Fig. 5 Mucinomics platform identified ovarian cancer patient similarities. A–E** Volcano plots representing five enrichment experiments from crude ovarian cancer patient ascites fluid. Ascites fluid samples were processed with the mucinomics workflow described in Figs. 1 and 2, including scoring with the mucin-domain candidacy algorithm. Strongly enriched proteins are labeled with gene names associated with specific proteins. MUC16 is labeled in purple. **F** Venn diagram comparing mucinomics results from five cell lines. Each sample is shown as a different color (red, orange, purple, green, and blue); 26 mucin proteins were enriched in all five samples. Graphic was generated using Intervene BattlePlot. **G** GO terms associated with mucin-domain glycoproteins. Enriched CC GO terms using DAVID[55,56] are shown with protein count indicated on the x-axis. Source data are in Supplementary Data 7 and 8. Significance testing was performed using a two-tailed t-test with 250 randomizations to correct for multiple comparisons, an FDR of 0.01, and an S0 value of 2 for all volcano plots.

provide O-glycopeptide identifications that include peptide sequence and the total glycan mass modification, though details about number of glycans or glycosite positions (and by extension, fine details about glycan structure) are usually inaccessible. Previous glycomic work suggests that some of these structures may include large, highly fucosylated and sialylated complex and hybrid N-glycans in addition to highly sialylated core-1 and-2 O-glycans with a smaller amount of sulfated core-2 O-glycan structures[62–64]. We collected only higher-energy collision dissociation (HCD) spectra through this study, limiting our ability to thoroughly characterize O-glycopeptides. Additionally, given that we performed in-gel tryptic digestion, it is unlikely that we were able to extract the intact mucin domains from many of our samples, nor were we able to fully characterize mucin domains of interest. Attempts to use StcE for in-gel digests resulted in limited digestion efficiency, and alternative methods to couple StcE proteolysis to this enrichment strategy are currently under investigation. Regardless, we searched our ascites data using O-Pair Search, a recently developed open-modification-centric glycoproteomic search algorithm that is particularly well-suited for the complex searches required of O-glycopeptide searches that consider large protein databases[65] (see Supplementary Data 9 for glycan databases used). Even though we could not capitalize on the site-localization capabilities of O-Pair Search, we identified several hundred glycopeptides in both the enriched and crude ascites samples; the total list of all glycopeptides identified is available in Supplementary Data 10 and 11.

Intriguingly, we discovered several O-glycopeptides on proteins that had previously uncharacterized mucin domains, as demonstrated in Fig. 6A. Here, the putative mucin domain is indicated by an orange box, annotated N-glycan sites are shown with green dots, and approximate location of the O-glycopeptides detected are shown using red dots. These proteins did not have any annotated O-glycosites in the SimpleCell dataset or in Uniprot,

thus these O-glycopeptides represent novel modifications on the mucin-domain glycoproteins. The presence of several identified O-glycopeptides in the regions assigned to be putative mucin domains by our mucin-domain candidacy algorithm also strengthens our claim that the proteins do, in fact, have mucin domains. Additionally, we detected a large number of glycopeptides from MUC16, which is a key step toward better structural definition of this important cancer antigen. The total glycan compositions for these peptides included N1, H1N1, N2, H1N2, N3, H1N1A1, H2N2, H1N2A1, H1N1A2, H2N2A1, and H2N2A2, where H is hexose, N is HexNAc, and A is Neu5Ac. The ratio of 138/144 in all of these cases was ~1, suggesting that the glycans are primarily core 1 (i.e., do not contain GlcNAc). Together, this would suggest that the compositions N2, H1N2, N3, H2N2, H1N2A1, H2N2A1, and H2N2A2 were multiply glycosylated peptides.

Next, we wanted to compare the glycoprotein sources of glycopeptides detected in the elution versus the crude cancer patient ascites fluid. As demonstrated in Fig. 6B, only 3% of glycopeptide spectral matches (glycoPSMs) originated from mucin-domain glycoprotein identifications in the unenriched ascites fluid, while 60% of glycopeptides from the elution came from mucin-domain glycoproteins. Further, 82% of all glycoPSMs in the elution were O-glycopeptides (rather than N-glycopeptides), compared to only 17% in ascites fluid (Fig. 6C). Supplementary Fig. 9 (data available in Data 9 and 10) shows the number of N- and O-glycopeptides detected in *n* number of experiments (where unique glycopeptide is defined as sequence peptide sequences and total mass combination), suggesting a significant biological variance in glycopeptide species between patients despite high protein-level overlap observed in Fig. 5F. We note that there is some level of ambiguity in glycopeptide identifications, given that 2 fucose residues may be assigned as a single sialic acid and vice versa. Regardless, to visualize the degree

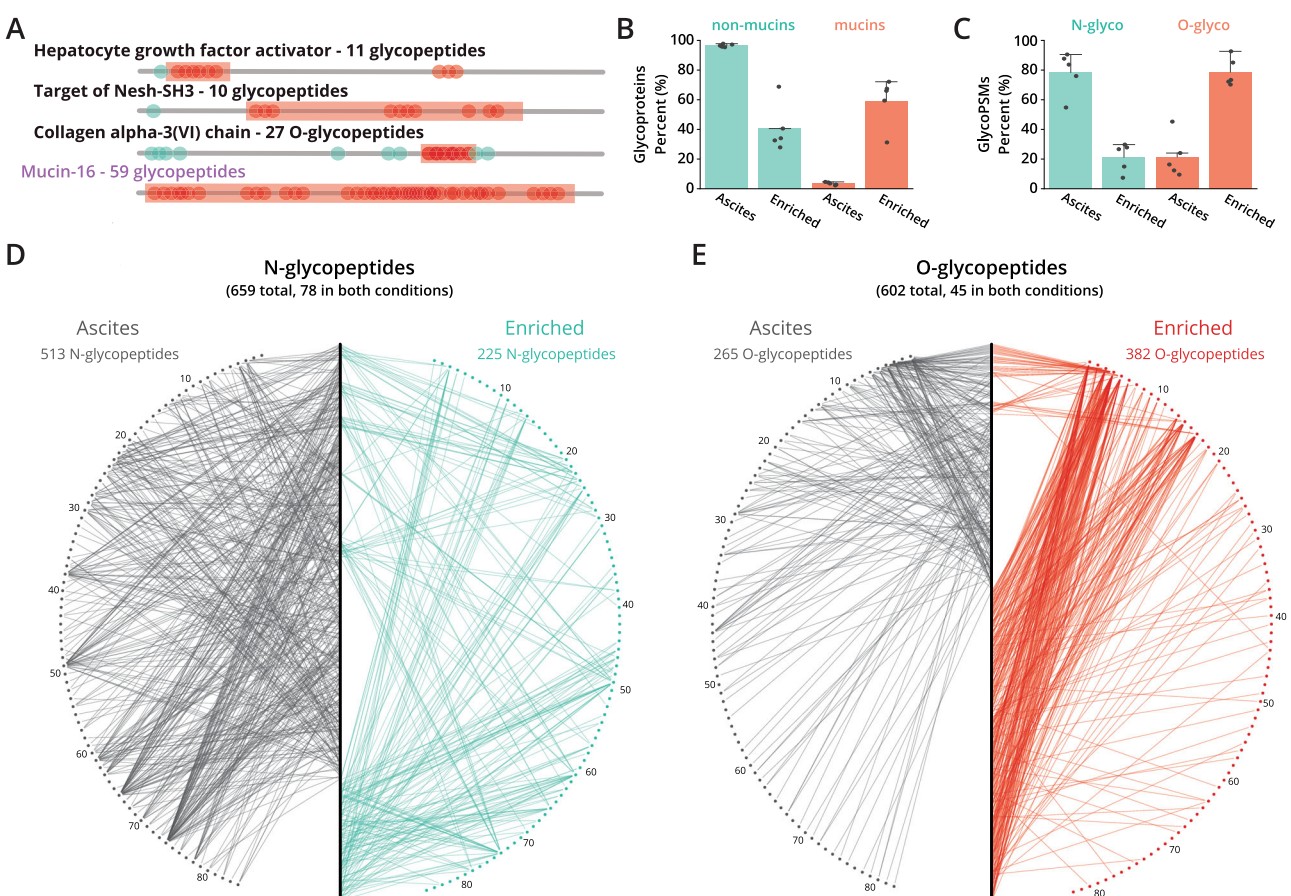

**Fig. 6 O-glycopeptides are highly abundant in the ascites enrichment. A** Mucin-domain glycoproteins harbor many O-glycosites. O-glycopeptides identified by O-Pair Search are shown in red circles in approximate locations on the protein backbone; proteins were annotated based on Uniprot assignments (N-glycan sites, green) and the mucin-domain candidacy algorithm (mucin domain, red). **B** O-glycopeptide identifications map to mucin-domain glycoproteins identified with the mucin candidate algorithm. Protein identifications from O-Pair glycopeptide searches were analyzed with the mucin-domain candidacy algorithm, and this bar graph shows the percentage of protein identifications from the O-Pair Search that were non-mucin proteins (green) and mucin-domain glycoproteins (red). **C** O-glycopeptide PSMs are higher in elution samples. The total number of N- and O-glycopeptides were summed for each sample. The percentage of total glyco peptide spectral matches (PSMs) that were either N- or O-glycopeptides are shown in green and red, respectively. In both panels **B** and **C**, values represent the average of the 5 patient samples, error bars show one standard deviation, and percentages for each of the five samples are shown as data points on the bar graph. Glycopeptide-glycan networks in panels **D** and **E** compare N- and O-glycopeptides, respectively, in ascites and enriched samples. For each glycopeptide-glycan network, unique glycopeptides (i.e., peptide sequence and total glycan mass) are organized vertically in the middle, and unique total glycan masses are nodes arranged in the outer semi-circles. The left and right semi-circles are mirror images of each other, showing the same glycan masses in the same order on either side, as indicated by numbers. The left side (gray) shows which glycan masses map to glycopeptides from ascites samples, and the right side (green or red) shows which glycan masses map to glycopeptides from enriched samples. These figures indicate which glycopeptides are shared or unique between the unenriched and enriched conditions and show that O-glycopeptides are detected more often than N-glycopeptides in StcE$^{E447D}$-enriched ascites fluid, the inverse of the non-enriched ascites fluid. Supplementary Data 11 and 12 provide total glycan mass compositions (outer nodes) identities and the unique glycopeptide list (middle nodes) for N-glycopeptide (**D**) and O-glycopeptide (**E**) networks, respectively. Source data are in Supplementary Data 10–13 and in Source Data.

of uniqueness/overlap between glycopeptides identified in ascites and enriched samples, we constructed glycopeptide-glycan networks shown in Fig. 6D, E, which are modified versions of previous protein-glycan visualizations introduced in Riley et al.[24] In these networks, unique glycopeptide identifications are arranged vertically as nodes in the middle of the network (black nodes in both panels). Unique glycan masses are then organized as nodes in the semi-circles on either side of the glycopeptide identifications, with each semi-circle representing the same glycan masses. In other words, gray nodes on the left of each network and color nodes on the right show the same glycan masses and are mirror images of each other. If glycan masses map to the same glycopeptide identifications, that means identifications are shared between the ascites (left, gray) and enriched (right, color) conditions. Otherwise, glycopeptide-glycan

connections that only appear on one side of the network are unique to that condition. In Fig. 6D, the majority of N-glycopeptides were identified in ascites rather than enriched samples, with relatively few N-glycopeptides mapping uniquely to the enriched samples. Conversely, Fig. 6E shows that the majority of O-glycopeptides were identified in the enriched samples, with the majority of those being unique to the enriched samples. Note, Fig. 6C denotes glycoPSMs whereas 6D and 6E are unique glycopeptide identifications. Slightly over 50% of all N-glycopeptide identifications in the enriched samples belonged to mucin-domain glycoproteins, while mucin-domain glycoproteins accounted for only ~15% of N-glycopeptide identifications from unenriched ascites fluid samples (Supplementary Fig. 9). Similarly, approximately two thirds (~66%) of all O-glycopeptide identifications in the enriched samples belonged to mucin-domain

glycoproteins, with only ~10% of O-glycopeptide identifications from unenriched ascites fluid samples deriving to mucin-domain glycoproteins (Supplementary Fig. 9). Detailed data underlying these glycopeptide-glycan networks are available in Supplementary Data 12 and 13. Overall, these data provide further evidence that we can selectively enrich mucin-domain glycoproteins with a concomitant increase in O-glycopeptide identifications.

## Discussion

A rapidly developing breadth of tools continues to shed light on glycobiology, which is historically understudied relative to other biomolecules. Mucin-domain glycoproteins represent one particularly challenging subset of the glycoproteome that remains poorly defined. Though canonical mucins are recognized as important contributors to health and disease, a "parts list" for the mucinome, i.e., a complete list of mucin-domain glycoproteins, remains elusive, even though the mucinome is poised to address many open questions in glycobiology.

Here, we used a point mutant of our mucin-selective protease, StcE$^{E447D}$, along with a mucin-domain candidacy algorithm to address this problem. We chose to build this candidacy algorithm on the hallmark mucin domain feature of serine and threonine O-glycosylation, as predicted by NetOGlyc4.0, while not focusing on other sequence characteristics such as proline frequency. While the enrichment feature of this mucinome workflow appears robust, we note that the mucin-domain candidacy algorithm is imperfect; yet, it serves a functional purpose for evaluating mucin-domain glycoprotein enrichments. Identification of mucin-domain glycoproteins more abundantly detected in cell lysates rather than the elution could also indicate that certain mucin domains remain under-glycosylated depending on cellular state or cell type, meaning our mucinomics approach could be used to screen the mucin status of proteins under a variety of conditions. Additionally, our mucin-domain candidacy algorithm could improve substantially from enhanced O-glycosite and mucin domain prediction tools. That said, prediction of mucin-type O-glycosites, much less mucin domains, remains challenging due to the complex regulation of O-glycosites by a poorly resolved family of glycosyltransferases. Future iterations could also explore other O-glycosite prediction algorithms beyond NetOGlyc4.0, such as ISOGlyP[66].

Though we have identified a subset of putative mucin-domain glycoproteins determined by the candidacy algorithm, we did not detect nearly 300 of these proteins. This can be likely be attributed to a number of reasons: first, we only explored 5 types of epithelial cancer cells; many other cancers and subsets of the same cancers are likely to express a different subset of mucin-domain glycoproteins. Also, we primarily used whole-cell lysates in this study, biasing toward membrane-tethered glycoproteins; given that mucin-domain glycoproteins can also exist as purely secreted biomolecules rather than membrane-tethered, it is possible that we missed a large number of mucin-domain glycoproteins only found in the secretome of cells. Further, it is entirely possible that the dense glycosylation in the mucin-domain glycoproteins renders them inaccessible to the in-gel digestion strategy used here. Current efforts are focused on optimizing the elution of the mucin-domain glycoproteins to enable in-solution digestion approaches. Finally, though previous experiments have suggested otherwise, it is possible that StcE enriches only a certain subset of mucin-domain glycoproteins from the samples. Interestingly, during the review process of this manuscript, Nason, Büll, et al. reported that the C-terminal domain of StcE can confer mucin-binding properties irrespective of the active site[67], meaning that the selectively of StcE$^{E447D}$ enrichments is not purely based on the O-glycosylated

TxT motif that dictates its protease activity. This generates interesting new directions to explore complexities of mucin binding harbored by catalytically inactive O-glycoprotease mutants.

Regardless, with this mucinomics platform, we enriched mucin-domain glycoproteins from several cancer-associated cell lines and crude ovarian cancer patient ascites fluid. We demonstrated high mucin overlap between ovarian cancer patients, and the enrichment strategy allowed us to detect hundreds of glycopeptides from the mucin proteins, with a substantial increase in O- over N-glycopeptides. We also identified many proteins previously unknown to contain a mucin domain, thus demonstrating the utility of this technique in discovering new mucin-domain glycoproteins. Future efforts will be devoted to expanding our patient cohort in order to determine whether the ovarian cancer mucinome can be used as a diagnostic and/or prognostic indicator.

Though this work represents a significant step forward in understanding mucin domains, several open questions remain. To begin, mucin domains are known to regulate interactions at cellular peripheries via biophysical effects and cell-to-cell interactions. However, these roles are likely extremely dynamic, and may depend on various glycan structures (alone or in combination), expression of the mucin domain, and the overall cellular milieu. Further, the role of an individual mucin domain is unlikely to be identical across all of the mucin-domain glycoproteins. Thus, future studies should be devoted to understanding the role that discrete mucin domains are playing in cellular function. We predict that these mucin domains will fall into subgroups with categorical roles in health and disease.

Additionally, while we have identified a large number of mucin-domain glycoproteins from cell lines and ascites fluid, many other mucin-domain glycoproteins are likely present on different cell types and in other indications. In particular, the immune cell mucinome is of incredible interest and may represent a class of new 'checkpoint inhibitors' with both glycan and peptide components to investigate[16]. Further, while we chose to focus our efforts on the cancer mucinome, several other mucinomes have yet to be studied in diseases known to involve dysregulated mucins. These mucinopathies include, but are not limited to, inflammatory bowel disease, cystic fibrosis, chronic obstructive pulmonary disease (COPD), Sjögren's syndrome, and dry mouth/eyes. Ultimately, we believe our mucinomics strategy will find utility in several settings and will prove to be an invaluable tool for glycobiologists and biochemists alike.

## Methods

**O-glycoprotease cloning, expression, and purification.** StcE and BT4244 were expressed as previously described[39,40]. Briefly, Natalie Strynadka (University of British Columbia) kindly provided the plasmid pET28b-StcE-Δ35-NHis[43]. Robert Hirt (Newcastle University) kindly provided the plasmid pRSETA-BT4244[33]. pET28b-StcE$^{E447D}$-Δ35-NHis and pRSETA-BT4244$^{E575A}$ were generated using the Q5 Site-Directed Mutagenesis Kit (New England Biolabs) with the following primers: StcE$^{E447D}$_for 5′-TCAGTCATGACGTTGGTCATAATTATG-3′, StcE$^{E447D}$_rev 5′-ACTCATTCCCCAATGTGG-3′, BT4244$^{E575A}$_for 5′-CCAG CTCATGCAATTGGCCATG-3′, and BT4244$^{E575A}$_rev 5′-TCCCCACGCGT TATCTTC-3′.

StcE$^{E447D}$ was expressed and purified as previously described[40]. BT4244$^{E575A}$ was expressed in BL21(DE3) E. coli (New England Biolabs) grown in Luria broth (LB) with 100 μg/mL ampicillin at 37 °C, 225 rpm. The culture was induced at OD 0.6–0.8 with 1 mM IPTG and grown overnight at 20 °C. Cell pellets were lysed in xTractor buffer (Clontech) and lysates were applied to 1 mL HisTrap HP columns (Cytiva Life Sciences) using an ÄKTA Pure FPLC. Columns were washed with 50 column volumes of 20 mM Tris-HCl, 100 mM NaCl, 15 mM imidazole, pH 8, and elution was performed with a linear gradient to 150 mM imidazole. For BT4244, fractions containing pure protein were concentrated using Amicon Ultra 10 kDa MWCO filters (Millipore Sigma), dialyzed into PBS, pH 7.4, and stored at −80 °C. BT4244$^{E575A}$ was further purified by size exclusion chromatography using a Superdex 200 Increase 10/300 GL column (Cytiva Life Sciences) in PBS, pH 7.4, and fractions containing pure protein were stored at −80 °C.

**Cell culture**. Cells were maintained at 37 °C and 5% $CO_2$. HeLa cells (ATCC CCL-2) were cultured in DMEM supplemented with 10% fetal bovine serum (FBS) and 1% penicillin/streptomycin (P/S). Capan-2 cells (ATCC HTB-80) were cultured in McCoy's 5a supplemented with 10% FBS and 1% P/S. K562 and SKBR3 cells (ATCC CRL-3344 and HTB-30, respectively) were cultured in RPMI supplemented with 10% FBS and 1% P/S. OVCAR-3 cells (ATCC HTB-161) were cultured in RPMI supplemented with 20% FBS, 0.01 mg/mL bovine insulin, and 1% P/S. To prepare lysate for pulldowns, cells plated in T75 flasks (Thermo Fisher Scientific) were grown until ~70% confluency, washed three times with DPBS, then lysed in 500 μL of RIPA buffer (Thermo Fisher Scientific) supplemented with EDTA-free protease inhibitor cocktail (Roche) and 0.1% benzonase (Millipore Sigma). Lysates were stored at −80 °C prior to pulldown.

**Bead derivatization**. An aliquot containing approximately 2 mg of StcE[E447D] (1 mL of 1.93 mg/mL) was added to 7–8 mg of POROS-AL beads, along with 1 μL of 80 mg/mL $NaCNBH_3$. The reaction proceeded overnight, with shaking, at 4 °C. After conjugation, the beads were washed three times with 500 μL of ultrapure water, spinning at 8500 rpm for 5 min each time. To cap all excess aldehyde sites on the beads, 200 μL of Tris-HCl with 1 μL of 80 mg/mL $NaCNBH_3$ was added to the beads. The reaction shook at room temperature for 2 h. Excess beads were stored at 4 °C for up to one month and were washed before each enrichment. Jacalin (Vector Laboratories, L-1150-25) derivatization was performed identically to the StcE[E447D] conjugation. For BT4244[E575A] conjugation, the enzyme concentration was 1.423 mg/mL, so 5.5 mg of POROS-AL beads was used. Otherwise, the conjugation and enrichment steps were identical.

**Enrichment of mucin-domain glycoproteins from cell lysates and ascites fluid**. Cell lysates were clarified by centrifuging for 20 min at 18,000 x g, and concentrations were determined using standard BCA assays. As per optimization experiments, the ideal ratio of lysate to beads (w/v) was determined to be 500 μg/100 μL, where 100 μL of the conjugated beads corresponded to 700 μg of beads in solution. The beads were pelleted at 8500 rpm for 5 min and the supernatant was removed. Then, 5 μL of 0.5 M EDTA and 500 μg of cell lysate was added to the beads and incubated at 4 °C overnight, with shaking. The reaction was performed six times, in tandem. After binding, the beads were spun at 8500 rpm for 5 min, and the supernatant was saved ("FT" or flow-through). Then, the beads were washed three times with 250 μL of PBS buffer containing 5 μL of 0.5 M EDTA. After the last wash, 32 μL of 4X protein loading buffer was added to the beads. For unenriched (control) samples, 30 μL of lysate was added to 10 μL of 4x protein loading buffer. All samples were then boiled at 95 °C for 5 min, spun for 2 min at 13,000 x g, and frozen for at least 1 h. The samples were then thawed and loaded onto 4–12% Criterion XT Bis-Tris precast gels (Bio-Rad), and run in 1x MOPS (Bio-Rad) for 90 minutes at 180 V. The total number of lanes for each experiment was 12, which included 6 control and 6 enriched lanes. After running, the lanes were stained using Bulldog Bio SafeStain and destained in ultrapure water. Eight bands were cut from each lane, giving a total of 96 slices per enrichment. The slices were frozen overnight at −80 °C.

For optimization and proof-of-principle purposes, only one replicate was performed, and all steps were run on a gel (FT, 3x washes, elution). Afterward, an anti-MUC16 Western blot was performed using anti-MUC16 antibody [X75] (Abcam, ab1107) at a dilution of 1:1000 and IRDye® 800CW Goat anti-Mouse IgG (LI-COR Biosciences, 926-32210) at a dilution of 1:25,000 according to manufacturer recommendations. Images (total protein, Western blot) were generated using an Odyssey CLx Near-Infrared Fluorescence Imaging System (LI-COR Biosciences).

Ascites from patients with gynecologic malignancies was collected with patient consent under an approved IRB protocol at from the Dept. of Obstetrics and Gynecology, Stanford Hospital. The study design and conduct complied with all relevant regulations regarding the use of human study participants and was in accordance with the criteria set by the Declaration of Helsinki. Ascites fluid was obtained from O.D. and V.K. and was de-identified prior to our handling. Samples were selected based on the amount of ascites available for the enrichment. BCA analysis revealed the average protein concentration to be 52 mg/mL, with a range of 33–64 mg/mL. In optimization experiments, the ideal ratio of lysate to beads (v/v) was determined to be 100 μL/100 μL, where 100 μL of the conjugated beads corresponded to 700 μg of beads in solution. Ascites was centrifuged at 4 °C at 18,000 x g for 20 min, and samples were removed from the supernatant. For control experiments, 6 μL of ascites was removed per lane for a total of 36 μL. Otherwise, the procedure was the same as above.

**In-gel digest and C18 clean-up for mass spectrometry**. All slices were thawed in 200 μL of ultrapure water (Pierce), followed by a rinse with 200 μL of acetonitrile (ACN, Fisher). Fresh 50 mM ammonium bicarbonate ("AmBic") was made, and samples were rinsed in 200 μL of AmBic for 20 min at RT. Afterward, samples were reduced using 5 mM dithiothreitol (DTT, Sigma) in AmBic for 35 min at 65 °C, with shaking, followed by alkylation using 50 mM iodoacetamide (IAA, Sigma) in AmBic for 30 min at RT, in the dark. Then, slices were rinsed once in AmBic, followed by two washes in fresh 50:50 AmBic:ACN for 10 min each. Slices were then dried in a vacuum concentrator and rehydrated with 0.1 μg of trypsin in

200 μL of AmBic and reacted overnight at 37 °C. The following day, samples were acidified with 2 μL of formic acid (FA, Thermo) and held at 37 °C for 45 min. The supernatant was discarded and 100 μL of 0.1% FA in 70% ACN was added to the slices for 30 min at 37 °C. The elution was collected, and the step was repeated once. The elution of adjacent slices was combined, for a total of 48 samples per enrichment. The resultant elution (400 μL) was dried in a vacuum concentrator).

All samples were subjected to desalting with a 96-well HyperSep C18 plate (Thermo). For all steps, solvent was added to the plate and centrifuged at 2013 × g in a Sorvall Legend RT. To begin, wells were wet with 150 μL of ACN followed by equilibration with 150 μL of 0.1% FA in ultrapure water ("solvent A"). Samples were reconstituted in 150 μL of solvent A and added to the plate three times. The wells were then washed three times with 150 μL of solvent A, followed by elution three times using 100 μL of 0.1% FA in 80% ACN ("solvent B"). The combined elution for each sample (48), totaling 300 μL, was taken to dryness in a vacuum concentrator. All samples were reconstituted in 7 μL of solvent A.

**Mass spectrometry**. Samples were analyzed by online nanoflow LC-MS/MS using an Orbitrap Fusion Tribrid mass spectrometer (Thermo Fisher Scientific) coupled to a Dionex Ultimate 3000 HPLC (Thermo Fisher Scientific) controlled by Xcalibur4.1 software. A portion of the sample was loaded via autosampler iso-cratically onto a C18 nano pre-column using 0.1% formic acid in water ("Solvent A"). For all cell lysate samples and enriched ascites fluid, 6.5 μL of sample was injected onto the column; for unenriched ascites fluid, 0.5–6.5 μL of sample was loaded onto the column as determined by peptide BCA (approximately 1 μg per sample). For pre-concentration and desalting, the column was washed with 0.1% formic acid in ACN and 0.1% formic acid in water ("loading pump solvent"). Subsequently, the C18 nano pre-column was switched in line with the C18 nano separation column (75 μm x 250 mm EASYSpray containing 2 μm C18 beads) for gradient elution. The column was held at 45 °C using a column heater in the EASY-Spray ionization source (Thermo Fisher Scientific). The samples were eluted at a constant flow rate of 0.3 μL/min using a 90 min gradient. The gradient profile was as follows (min:% solvent B, 2% formic acid in acetonitrile) 0:3, 3:5, 93:25, 103:35, 104:90, 109:90, 110:3, 140:3. The instrument method used an MS1 resolution of 60,000 at FWHM 200 m/z, an AGC target of 3e5, and a mass range from 350 to 1,500 m/z. Dynamic exclusion was enabled with a repeat count of 3, repeat duration of 10 s, exclusion duration of 10 s. Only charge states 2–6 were selected for fragmentation. MS2s were generated at top speed for 3 s. HCD was performed on all selected precursor masses with the following parameters: isolation window of 2 m/z, 30% collision energy, orbitrap detection (resolution of 30,000), and an AGC target of 1e4 ions.

**Mucin-domain candidacy algorithm**. To build the mucin-domain candidacy algorithm, the entire human proteome was first downloaded from Uniprot (20,365 entries) and parsed into FASTA files containing 150 entries each (a total of 136 files). Each file was individually uploaded to the NetOGlyc4.0 Server (http://www.cbs.dtu.dk/services/NetOGlyc/) for O-glycosite prediction[51]. NetOGlyc4.0 results were saved as.csv files for further processing, with 20,121 entries returning usable output. Those without a NetOGlyc4.0 output received a Mucin Score of NaN in the supplemental datafiles, which differs from a score of 0 that can be calculated through the description below. Cellular component (CC) GO terms for the human proteome were also downloaded from Uniprot, and phosphosite annotations were downloaded from Uniprot and PhosphoSitePlus[52,53]. Predictions from NetO-Glyc4.0 were then screened for known phosphosites, and any overlap in phosphosites with predicted O-GalNAc sites resulted in removal of the predicted O-GalNAc site from consideration. To annotate proteins as extracellular, secreted, and/or transmembrane, cellular component localization terms from Uniprot were checked for each protein entry. A protein was annotated as "extracellular" if its CC GO terms contained the phrases "Cell Membrane", "Cell membrane", "pass membrane protein", "Secreted", "extracellular", or "Extracellular". Proteins also received the "extracellular" distinction if they contained GO accessions of 0005887, 0016021, or 0005576. Because many proteins have multiple locations, "extra-cellular" proteins were further denoted as "exclusively extracellular" if their GO term lists did NOT include "Mitochondrion", "Cyto", "cyto", "Nucl", or "cyto-plasmic side". Next, predicted O-glycosites were iterated over to determine if a given protein would pass our "mucin test", which consisted of two calculations. First, we required a protein to have at least nine predicted O-glycosites within a 50-residue region. If a protein qualified for this benchmark, we applied our "12% rule" to determine the number of residues that separated any two given O-glycosites within this 50-residue region. The 12% rule applied to a 50-residue region meant that fewer than 6 residues could separate any given pair of O-glycosites. Both the "9 sites within 50 residues" metric and the "12% rule" were derived through hand annotation of known and thoroughly studied mucins mostly curated by the Mucin Biology Group (http://www.medkem.gu.se/mucinbiology/databases/db/Mucin-human-2015.htm)[68,69]. Although this could be considered both too stringent or too relaxed depending on perspective, empirical testing showed these rules (in con-junction with the other metrics discussed) to be reasonably reliable in properly annotating known mucin domains. Exploration of these "mucin test" metrics in particular is an interesting area for future studies looking to employ a mucin-domain candidacy algorithm. Finally, a threonine to serine ratio (T/S-ratio) was calculated for the predicted O-glycosites, mainly as a metric to discriminate

O-GalNAc sites (slight threonine preference) from phosphosites (slight serine preference) due to the proclivity of NetOGlyc4.0 to predict dense regions of O-GalNAc sites in what are actually intracellular phosphorylation domains. Note, these preferences are based on empirical observations. If the number of serines and threonines were both greater than zero, the T/S-ratio was calculated by taking the number of threonines and dividing by the number of serines. If the number of threonines was > 0 but the number of serines was 0, the T/S-ratio was assigned a value of 2. Otherwise, the T/S-ratio was set at 0. With all of these determinations made, we then generated a Mucin Score. First, an integer score was calculated. If a protein was annotated as "extracellular" and passed the "mucin test", it received an integer score of 1, while proteins "exclusively extracellular" and passing the mucin test received an integer score of 2. Integer scores were then augmented by 1 point if the predicted number of O-glycosites was greater than the number of annotated phosphosites. Finally, the integer score was multiplied by the T/S-ratio to generate the Mucin Score. This process was completed for all proteins in the human proteome that had predictions returned from NetOGlyc4.0 (~20,191 entries). Mucin Scores were used to determine confidence that a protein contained a mucin domain, including valuations of high confidence (Mucin Score > 2), medium confidence (2 > Mucin Score > 1.5), low confidence (1.5 > Mucin Score > 1.2), and non-mucin (Mucin Score < 1.2). This annotation was determined manually by assessing all of the factors above. For regions of 50 amino acids identified as putative mucin domains through this analysis, approximately 90% of them mapped to a single exon, which was evaluated manually using the neXtProt knowledgebase[70].

**Unmodified peptide MS data analysis (MaxQuant)**. Raw data were processed using MaxQuant version 1.6.3.4, and tandem mass spectra were searched with the Andromeda search algorithm. Oxidation of methionine and protein N-terminal acetylation were specified as variable modifications, while carbamidomethylation of cysteine was set as a fixed modification. A precursor ion search tolerance of 20 ppm and a product ion mass tolerance of 20 ppm were used for searches, and two missed cleavages were allowed for full trypsin specificity. Peptide spectral matches were made against a target-decoy human reference proteome database downloaded from Uniprot. FBS contamination was not examined for the lysate samples. Peptides were filtered to a 1% FDR and a 1% protein FDR was applied according to the target-decoy method. Proteins were identified and quantified using at least one peptide (razor + unique), where razor peptide is defined as a non-unique peptide assigned to the protein group with the most other peptides (Occam's razor principle). Proteins were quantified and normalized using MaxLFQ[71] with a label-free quantification (LFQ) minimum ratio count of 1. LFQ intensities were calculated using the match between runs feature, and MS/MS spectra were required for LFQ comparisons. For quantitative Article comparisons, protein intensity values were log2-transformed before further analysis, and missing values were imputed from a normal distribution with width 0.3 and downshift value of 1.8 (that is, default values) using the Perseus software suite[48]. A Boolean value "IsAMucin" was also appended to each protein, with the value set as true if the Mucin Score was greater than 1. Mucin Scores and IsAMucin were input manually into MQ 'protein groups' txt files for manipulation in Perseus. Significance testing was performed in Perseus using a two-tailed t-test with 250 randomizations to correct for multiple comparisons, an FDR of 0.01, and an S0 value of 2 (all volcano plots), or in Microsoft Excel using a two-tailed t-test with heteroscedastic variance. We kept the standard Perseus column headers from these analyses, with "Significant" showing a "+" for proteins calculated as significant based on the t-test performed, -log(P-value) providing the y-axis value of the volcano plots that shows the log-transformed value of the t-test p-value, and "Difference" indicating the log2 fold change between the conditions (e.g., elute and lysate). Proteins were sorted by their Mucin Score and highlighted in red if the score was higher than 2 ("high probability mucin"), orange if between 2–1.5 ("medium probability mucin", and yellow if between 1.5 and 1 ("low probability mucin"). Upset plots and the 5-sample Venn diagram (Figs. 4A and 5F, respectively) were generated using the Intervene Shiny app (https://intervene.shinyapps.io/intervene/)[72]. GO term enrichments were performed using DAVID[55,56], with the human proteome as a background.

**Glycopeptide MS data analysis (O-Pair Search)**. For glycopeptide analysis, samples were loaded into MetaMorpheus in groups of 8, related to one individual replicate (e.g. "Lysate 1" slice 1–8)[65,73]. The human proteome was loaded into the database (downloaded from Uniprot June, 2016), and a "Glyco" search task was selected. For each group of 8 raw files, an N- and an O- glyco search was performed separately. Parameters for the O-Glycopeptide Search were as follows: O-glycan database "Oglycan.gdb" (the default 12-glycan database[65,74], keep top 50 candidates, Dissociation type "HCD" and child scan "null", 4 maximum Oglycan allowed, with OxoniumIonFit on. For the N-Glycopeptide Search, all parameters were the same except the "NGlycan182.gdb" database was used. These glycan databases are available in Supplementary Data 9. For general peptide parameters, the following features were used: tryptic cleavage, maximum missed 2 cleavages, maximum 2 modifications per peptide, with a peptide length of 5–60. Precursor mass tolerance was set to 10 ppm, product mass tolerance at 20, with a minimum score allowed of 3. Finally, carbamidomethyl Cys was set as a fixed modification, whereas oxidation of Met was set as a variable modification. All glycopeptide hits

were filtered to have a Q value of less than 0.01 and all decoy hits were removed. In O-glycopeptide searches, any peptides that had the "N-glyco sequon" as "TRUE" were also removed. Bar graphs in Fig. 6B, C were made using OriginPro 2022 and show the average value of the five data points shown indicated along with standard deviations. The glycopeptide-glycan networks in Fig. 6D, E were created in R 3.5.1 using the igraph library[75].

**Reporting summary**. Further information on research design is available in the Nature Research Reporting Summary linked to this article.

## Data availability

The raw mass spectrometry data generated in this study have been deposited in the PRIDE database[76] under accession code PXD024995. The SimpleCell dataset from Clausen and colleagues was obtained from Steentoft et al.[51] (Supplemental Table 2 in that publication). The proteomics data generated from the mucinome enrichments of cell lysates and ascites fluid, the outputs from the mucin candidacy algorithm, the glycan databases used for glycopeptide searches, the glycoproteomics data generated from the mucinome enrichments of ascites fluid, data to make the N- and O-glycopeptide networks, and data to recreate figures are provided in the Supplementary Data files as indicated in the text. Source data are provided with this paper.

## Code availability

Code for the mucinome candidacy algorithm is available as Supplementary Software 1.

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

## Acknowledgements

We thank Natalie Strynadka (University of British Columbia) and Robert Hirt (Newcastle University) for their gifts of the StcE and BT4244 expression plasmids, respectively. We also thank Jessica Stark and Rishikesh Kulkarni for helpful discussions. This work was supported, in part, by National Cancer Institute Grant R01CA200423 (to C.R.B.) and the Stanford Women's Cancer Center Innovation Award 123-0040-100-WCHGC (to C.R.B. and O.D.). S.A.M. was supported by a National Institute of General Medical Sciences F32 Postdoctoral Fellowship (F32-GM126663-01) and is currently supported by the Yale Science Development Fund. N.M.R. was funded through an NIH Predoctoral to Post-doctoral Transition Award (Grant K00 CA212454). D.J.S. was supported by a National Science Foundation Graduate Research Fellowship and Stanford Graduate Fellowship. K.P. was supported by a National Science Foundation Graduate Research Fellowship, a Stanford Graduate Fellowship, and the Stanford Chemistry, Engineering & Medicine for Human Health (ChEM-H) Chemistry/Biology Interface Predoctoral Training Program. Raw and processed data are available through the PRIDE database, accession PXD024995.

## Author contributions

S.A.M. and C.R.B. designed research; S.A.M., N.M.R., D.J.S., and K.P. performed research; V.K. and O.D. contributed human clinical samples; S.A.M., N.M.R., D.J.S., and K.P. analyzed data. S.A.M., N.M.R., and C.R.B. wrote the paper with input from all authors.

## Competing interests

S.A.M., D.J.S., K.P., and C.R.B. are coinventors on a Stanford nonprovisional utility patent application that has been filed and is pending in the US (number US20220003777) related to the use of inactive mucinases to enrich mucin-domain glycoproteins. C.R.B. is a co-founder and Scientific Advisory Board member of Lycia Therapeutics, Palleon Pharmaceuticals, Enable Bioscience, Redwood Biosciences (a subsidiary of Catalent), and Inter-Venn Biosciences, and a member of the Board of Directors of Eli Lilly & Company. O.D. has participated in advisory boards for Tesaro, Merck, and Geneos. O.D. is a speaker for Tesaro and AstraZeneca. The remaining authors declare no competing interests.
