## [Peer Review File · Nature Communications]

REVIEWER COMMENTS

Reviewer #1 (Remarks to the Author):

In their manuscript „Revealing the human mucinome” Malaker et al. described an impressive approach to enrich mucin domain glycoproteins from cell cultures and ovarian cancer patient ascites fluids. The authors exploited the mucin domain binding capacity of a catalytically inactive mucinase followed by separation, in-gel digestion and LC-MS/MS analysis of the isolated protein samples. Comparison to untreated controls using label-free protein quantification showed the enrichment of 20-50 mucin domain glycoprotein candidates in the different samples. Furthermore, the authors developed an elaborate algorithm to predict mucin domains in glycoproteins and used this algorithm to corroborate the enrichment of potential mucins. Finally, the authors also identified glycopeptides to demonstrate the mucin-like glycosylation pattern of glycoproteins enriched from patient ascites fluid samples. Considering the relevance of mucins in disease pathologies and immune responses and currently limited knowledge on the mucinome this manuscript is timely and the novelties justify publication with minor modifications.

Although it might seem a minor technicality, my major concern is the reliability of data interpretation used for the label free quantification. In the Methods sections the authors wrote that „precursor ion search tolerance of 20 ppm and a product ion mass tolerance of 0.3 Da were used for searches”. As the Orbitrap analyzer was used for MS2 data acquisition the mass errors should be well below 20 ppm. Why did the authors use this high product ion mass tolerance? 0.05 Da should have been still very permissive if relative mass accuracy could not be defined. If there was no second round of peptide identification with stricter parameters the reliability of the protein identifications is disputable. If this is the case the authors should reanalyze the data (or at least demonstrate on one particular dataset that the results are very similar with strict and loose search parameters). I did see that the authors applied FDR thresholds but the mass accuracies also affect the search space and hence the peptide FDR values.

My minor questions and comments:

1. Canonical mucins are usually very difficult to digest due to heavy O-glycosylation of the mucin domains. Here, the authors could identify 59 O-glycopeptides from Mucin16. Were these peptides multiply glycosylated? (Unfortunately, I could not access the relevant data in the repository. In fact, access to data in the Pride repository is very frustrating. One cannot search for specific files, max 40 files/page can be viewed at a time and any arrangement is possible on the page level. Please include the list of O-glycopeptide identifications as supplementary material along with all necessary data to enable manual assessment of glycopeptide IDs from raw data deposited.)

2. It would be interesting to see the mucin scores for the novel mucins described in the results. Please include these in the text and also in Suppl Tables 1-3.
3. On p9 the authors describe their preliminary results using a mutant mucinase presumably binding mucin domain glycoproteins decorated only with truncated O-glycan structures (T and Tn antigens). The authors show that only a limited number of potential mucins are enriched significantly and claim that this is in good agreement with the expected non-truncated O-glycosylation pattern of HeLa glycoproteins. However, without demonstrating that the enriched mucins indeed display truncated glycans these results are somewhat unreliable. Furthermore, the portion of the enriched mucins in this particular sample (7/37, ~20%) is similar to the portion of depleted mucins from cell lines using the mutant SctE (30-50%), therefore it could be argued that the 7 enriched mucins are kind of „overall FDR mucins” similar to the mucins depleted in enriched samples using the SctE mutant. The findings described by the authors could be a starting point for another study aiming the isolation and site-specific characterization of truncated cancer-specific glycoforms, but does not add to the story of the current paper in my opinion, I suggest removing this part.
4. Figure 6D is difficult to interpret. First, the same width for both N- and O-glycopeptide IDs visually suggests that the same number of N- and O-glycopeptide identifications were obtained. Second, it is impossible to assess the differences between the enriched/ascites samples. It is evident that N-glycopeptides are more prevalent in the ascites samples. On the other hand it seems that there is a higher number of O-glycopeptide IDs in ascites compared to enriched sample in OC235. Please include total number of glycopeptide IDs for each sample.
5. Also, it would be interesting to see the overlap between O-glycopeptides (sequence+net glycan composition) identified from the different enriched ascites samples. Is there a set of peptides that were identified in all samples? (Figure 6D suggests significant biological variance.) Could these be used a biomarker panel to diagnose ovarian cancer? Are there any glycoproteomic data available for other pathologies resulting in ascites such as cirrhosis or other cancer types?
6. What was the overlap of enriched mucin domain glycoproteins and glycopeptides identified from the ascites samples (what portion of the O-glycopeptides represented the enriched mucin domain glycoproteins)?
7. What is the appr. protein concentration in ascites fluid and to what extent might it vary between patients? Was there any total protein amount estimation of the ascites fluid samples? If yes, please include. MS method description indicates that peptide-level BCA yielded highly varying total protein amount of ascites fluid samples.
8. Please include the O- and N-glycan databases considered for O-Pair searches in the Supplementary. How did the authors decided which 12 O-glycans to consider?
9. Mucin scoring indicates that there are some 350 human mucin domain glycoproteins. The current study identified ~50 of these. Could the authors comment on the „missing” 300? Are there any info on the expression level and/or trypsin accessibility of the „missing” proteins?

Minor comments:

1. In Figure 2 caption and on page 7 the authors say „approximately 330 proteins contain a mucin domain by our estimate (score > 1)” – in suppl dataset 1 there are exactly 350 proteins with mucin score >1. Please correct.
2. Page 8: „Several of the proteins were previously known to have a mucin domain”... as the authors discuss 6 proteins altogether, „several” does not seem to be the appropriate wording here. Please rephrase.
3. In suppl dataset 2, „Capan2” tab need to be revisited. In the „IsaMucin” column many proteins are not classified, one is indicated as „UD14_HUMAN”.
4. Please elaborate on „mucin score” values. How is „NaN” and „0” different?
5. In the cell line experiments, did the authors check for FBS contamination in the samples? In the methods section it is described that only the Uniprot/human database was considered in the database searches.
6. Results section, p4: „Through a straightforward reductive amidation approach” – the protein is immobilized via an amine linkage, so this is a reductive amination not amidation, please correct.
7. Methods section: „Ascites fluid was obtained from O.D. and V.K.” – please elaborate (patients? hospital administrators? etc). „Samples were selected based on their grouping in ref 28” again, please clarify.
8. Methods section, in-gel digestion: it is written that following tryptic digestion and acidification the supernatant of the gels were discarded and only the ACN extract was kept for downstream LC-MS/MS. In my experience, the aqueous „extract” contains a large portion of the tryptic peptides, therefore it would be interesting to know why these samples were excluded from analysis. Please clarify.
9. LC-MS: solvent B, 2% formic acid in acetonitrile – is that correct?
10. LC-MS: „MS1 resolution of 60,000 at FWHM 400 m/z” – the resolution is defined at m/z 200 for the instrument used in the experiments.
11. It is a bit difficult to interpret data included in the Suppl datasets. First column: „significant”, second column: (-LOG(P-value)), third column: Difference - what these columns mean? How are these values calculated, what cut-offs were used? Please include legends to explain data in the Supplementary dataset tables.

Reviewer #2 (Remarks to the Author):

The Bertozzi group has used inactive StcE lectin to purify and analyze proteins and tryptic peptides from cell lines and ascites. By this they characterized a number of proteins in a few cell lines and in

ovarian cancer ascites previously not known to be O-glycosylated as listed in the Simple Cell dataset. They have analyzed the samples by mass spectrometry and the hits by various criteria and bioinformatics.

This is a potentially interesting publication, but also contain some mistakes and over estimations. To use lectins to purify glycoproteins has been used for many years where some plant lectins, like Jacalin, was thought to be more specific for mucin domains and thus the approach is not new. Although StcE clearly binds clustered O-glycans as found in mucin domains, it is highly unlikely that it will bind all such domains as binding will depend on the glycan structures and their arrangements. Thus we need to have better knowledge of the specificity of the StcE binding to know which mucin domains that are bound. The specificity of StcE is good and the approach is good, but the text is claiming that all mucin domains will be enriched is likely not the case. However, the authors have isolated a subset of mucin-domain containing proteins of which some where not described as having a mucin domain before.

The purified proteins were separated on SDS-PAGE and all components cut out and analyzed by mass spectrometry. The enriched proteins were identified after trypsin cleavage by sequences outside of the mucin domains as these will be lost during the preparation and not possible to identify within the mass range studied. This has to be explained. Most important, the authors have to present the mucin domain peptide sequences of the identified proteins allowing the reader to interpret their predictions of mucin domains.

This also means that the identification is not very different from other ways of mining DNA/protein sequences. Typical for mucin domain sequences are their lack of sequence conservation over evolution, long sequences, only proteins with signal sequence or transmembrane domains, and the high abundance of Ser and Thr amino acids interrupted by Pro. This rendered the name 'PTS sequences' that is preferred to the 'VNTR sequence' as many mucin domains lack an identifiable repeat structure. The characteristics for a PTS has been used for mining genomic databases by the Samuelsson group (Lang, T., Alexandersson, M., Hansson, G. C., and Samuelsson, T. Bioinformatic identification of polymerizing and transmembrane mucins in the puffer fish *Fugu rubripes*. *Glycobiology* 14, 521-527. 2004; Lang, T., Hansson, G. C., and Samuelsson, T. Gel-forming mucins appeared early in metazoan evolution. *Proc.Natl.Acad.Sci.USA* 104, 16209-16214. 2007). This is not discussed despite that the criteria for a mucin domain length and Ser/Thr content is similar to that used before.

They relate their information to the Simple Cell database but this is partly irrelevant (Tables S1-S2) as this approach is excellent for identifying single O-glycans, but poor for larger clustered O-glycans as in mucin domains. The NetOGlyc4 tool also has limitations as this is not taking into account that there are 20 different peptidyl-GalNAc transferases with different specificities as pointed out by Gerken et al (Daniel, Earnest James Paul, Las Rivas, Matilde, Lira-Navarrete, Erandi, Garcia-Garcia,

Ana, Hurtado-Guerrero, Ramon, Clausen, Henrik, and Gerken, Thomas A. Ser and Thr Acceptor Preferences of the GalNAc-Ts Vary Among Isoenzymes to Modulate Mucin Type O-Glycosylation. *Glycobiology* . 4-18-2020)

Another major issue for the manuscript is nomenclature. There is a set of human glycoproteins whose genes has been named mucins. This is a heterogeneous group of glycoproteins, but it does not help that the authors starts to call also additional proteins mucins. Stick with the nomenclature of 'mucin domain containing proteins' also for the figure 2 and text. The authors define what a mucin domain is by stating that there should be >50 amino acid residues with predicted 10 O-glycan sites. All mucin domains so far are found in a single exon, something that should be analyzed for the new mucin domains.

The use of the word 'mucinome' is acceptable, but mucinases is imprecise as it is often used for bacterial O-glycopeptidases and O-glycohydrolases. How about 'mucin-domain peptidase'? I should avoid mucinopathologies, what is this? Mucinomics is also imprecise and suggest a method for identifying all mucin domains, something that is not possible today where we know that some mucin domains have molecular masses in the MDa range.

Page 3

- The abstract claim to identify mucins, should be removed.
- Not all mucin domins occur as tandem repeats – most of the mucin domains described here do not have tandem repeats.

Page4

- StcE does not cleave all mucin domains and is thus not a pan-mucinase.

Page 4

- Figure 1: Most mucin domain glycoproteins have their mucin domain close to the membrane to generate a stalk, like the LDI receptor
- The authors have not developed a mucin candidacy algorithm, maybe a mucin domain candidacy algorithm
- It is unclear why the authors chose to analyze for phosphosites as transmembrane proteins might have intracellular phosphorylation and extracellular O-GalNAc. Should it not be enough to analyze for secreted or transmembrane proteins that is done in next step?

Page 5

- Mucin domains also has to contain a substantial frequency of Pro, why was this nt considered?
- Fig. 2 Mucin should be replaced by mucin domain

Page 8

- Remove 'pan-mucinase'
- Figure 4 is difficult to understand. Why the left hand subfigure? The presence of which of the 6 proteins shown are present in each of the 1-5 bars? Why are there 3 bars for the 6 proteins?? Replace Mucins with mucin containing glycoproteins.
- The approach is mucin domain enrichment approach and nothing else.

Page 10

- The tryptic fragments analyzed will not reveal the large classical mucin domains as these are too large and lost during preparation. Should be discussed instead of the ETD vs. HCD.

Page 11

- Fig. 6. Replace mucin.

The word mucin should be replaced by mucin domain in almost all instances throughout the text.

Reviewer #3 (Remarks to the Author):

The manuscript from Malaker & Riley et al. is a well written manuscript that uses a novel approach to develop a tool that can be used to reveal an under-appreciated part of the proteome: the "mucinome". I appreciate the general approach to this problem, but I believe that there are a few points that need to be clarified in this manuscript so that the full potential of this work can be realised. If it was possible to analyse the data to give more indications as to what the exact patterns are that the inactivated mucinases/glycopeptidases are enriching for, I think this would improve this manuscript. For example, do they enrich for a particular O-linked glycan on the proteins, are

sequence motifs (e.g. TxT) recognised by the inactivated mucinases, and so enriched for in the resultant proteins?

The authors have decided upon the presence of mucin domains (defined essentially as a minimal density and length of glycosylated regions) as the major factor that contributes to this enrichment. I would like to see a bit more information about alternative hypotheses (all examined bioinformatically) to explain the enrichment, especially taking into account what is known about the specificities of the active mucinases. While the data as it stands may not allow for insight into which elaborated/truncated glycans are present, information about potential density of glycosylation and motifs could be extracted, and compared to the mucin-domain candidacy algorithm.

Key results

- The mucinome is a subset of the human proteome that is relatively poorly characterised, and can be revealed using mass spectrometry and selective enrichment using inactivated mucinases
- The mucinome can be enriched for using catalytically inactivated mucinases (StcE & BT4244).
- They enrich mucins because there are more proteins classified as mucin/mucin-like in the enriched portions.
- The classification of mucin and mucin-like domain containing proteins uses a mucin-domain candidacy algorithm.
- The enrichment can define key molecular signatures of cancer in-vitro / ex vivo.

Validity

Nomenclature / naming:

- Nomenclature is always a difficult discussion to have because the reasoning behind preferences is often highly subjective (highly dependent on experience and research focus). However, I think the usage of “mucin” as opposed to “mucin-like” to describe regions of dense O-GalNAc glycosylation has the potential to introduce confusion. Mucins have conventionally been defined as proteins that contain tandem-repeats with a high density of PTS, and the repeating region is integral to protein identity. Complicating this, there are proteins called mucins, that don't have repeats (e.g. MUC13), or proteins that have PTS repeats but aren't mucins (e.g. GP1Ba or CSPG). Furthermore, a number of proteins have “mucin-like” domains (e.g. DAG1 and KIAA1549), characterised by large amounts of O-

glycosylation (in both these cases, they carry O-Mannose and O-GalNAc residues). Finally, many proteins have shorter regions that have high densities of glycosylation, often found as linker regions between domains, stems on Type 1 or 2 proteins or on N- and C- termini of proteins (DOI: 10.1007/978-4-431-54841-6_63). All these can have high density of glycosylation in longer stretches, but they're not all mucins.

Functionally, the mucins fulfil specific roles within their respective niches, with function derived from not only the high density of glycosylation, but likely also impacted by the tandem repeats on the protein. Overloading the meaning of the mucin to include proteins that are likely much more functionally diverse will conflate the functions that are commonly associated with mucins with any region that has a high density of glycosylation. The term "mucin-like" does not have this problem as it has been historically used to describe regions carrying large amounts of O-GalNAc glycosylation, but is an awkward term to use as an 'ome.

If the authors have other suggestions for a specific umbrella term to describe these regions of high density glycosylation (which could be further sub-divided in to mucin, mucin-like and short mucin-like) this would sidestep these issues.

Validation of enrichment:

- The inactivated mucinases enrich a subset proteins, but it is not known whether they specifically enrich for mucins or mucin-like proteins. The authors suggest that the presence of mucin domains on proteins lead to the enrichment and thus categorise proteins into groups depending on the likelihood that they have mucin domains. The analysis assumes that this categorisation system that they use is the most appropriate, but any comparison of using this categorisation, as compared to other potentially simpler systems is not provided. There are a few simple categorisations that could be relevant to compare the mucin-domain candidacy algorithm against.

From the previous literature (Malaker et al. 2019, PNAS, Shon et al. 2020, PNAS), the specificity of the mucinases StcE is primarily towards a TxT motif (with glycosylation), while BT4244 prefers a glycosylated threonine. Two metrics could be derived: the counts (abundance) of TxT motifs, and the counts of TxT motifs that are predicted to be glycosylated (using e.g. NetOGlyc). Are there any correlations between the number of TxT motifs and the resultant quantitation of protein enrichment?

Another metric that could be used is the total relative / absolute length of regions of high density of glycosylation on proteins. If you look at MUC4 and MUC16, you can expect that approximately 6/10

amino acids are possibly glycosylated in their repeat regions (and 5/10 amino acids for MUC1). If you run a sliding window across the protein sequence, you can calculate these densities for proteins. Similarly to the mucin candidacy algorithm suggested in the manuscript, these calculations can be limited to extracellular regions.

- The authors do not comment on how many (or what proportion of) proteins in the enriched portion do not have mucin-like properties. Quantification of this will provide for a confidence that the inactivated mucinases specifically enrich for mucin/mucin-like domains.
- For those non-mucin proteins that are reproducibly enriched, and lack a TxT motif, do the authors have any theories as to why these proteins are enriched? Is this an indication of a broader specificity of the enrichment behaviour of the inactivated mucinase? I have taken a brief look at the supplemental tables, and it looks a lot like there are many nucleocytoplasmic proteins that are enriched. If this is the case, then maybe these proteins could be recoloured/de-emphasised in the volcano plots as they are unlikely to be directly enriched by the mucinases? (Unless somehow the mucinases enrich for O-GlcNAc too).
- Figures 2A and C could be reworked so that it is easier to benchmark the performance of the mucin candidacy algorithm. It would be useful to be able to evaluate whether the algorithm can successfully identify the known enriched mucins (these points remain green in Figure 2C, so you cannot evaluate the confidence that these proteins are mucins). Secondly, how many proteins are both enriched, and not identified as mucin-domain proteins? In Figure 2A, clearly there are many more proteins not identified as mucin than are mucin, but it is harder to derive this information from Figure 2C. If both the mucin-domain candidacy algorithm works, and the inactivated StcE enriches mucin/mucin-like domain containing proteins, I would expect that there are more proteins predicted to contain mucin-like domains in the enriched portion than proteins not predicted to be mucins.

If this is not the case (more non-mucin proteins than predicted mucins in enriched portion) than I would consider that the enrichment possibly enriches for some other arrangement of glycosylated sites than the arrangements selected for by the domain candidacy algorithm.

Specific questions about the mucin candidacy algorithm:

- While this phenomenon hasn't been systematically investigated, it is known that extracellular phosphorylation overlaps with O-GalNAc sites. This is most well-known on FGF23 (by FAM20C), but can also be seen on CD55/DAF, so removing GalNAc/phosphorylation site overlaps isn't 100% correct.

- The “12% rule” is defined as 12% of entire mucin domain, but the way that the mucin domain is derived isn’t clarified (i.e., what decides the length of the mucin domain?). The length of the mucin domain will have an impact on whether this criterion is passed or not.

- Why add extra points for proteoglycan names (and mucin names)? Wouldn’t this be better served by adding a whitelist for proteins that have the desired properties (e.g., CSPG) and can be classified as containing mucin-domains? Using names in this algorithm is fragile because protein names are subject to change, and as mentioned earlier, some proteins with mucin names aren’t technically mucins.

General comments:

- There is a lot of other data available other than that published by Steentoft (SimpleCell) in 2013, and that data found in UniProt. Much more data has been published on the O-glycoproteome in the last 8 years, and includes data published by the Clausen lab, various serum O-glycoproteomic data sets, IsoTag data (Woo et al. 2017), and data from OperATOR enrichment (DOI: 10.15252/msb.20188486) which in principle also uses a mucinase to enrich (albeit for glycopeptides). It is likely that some of the novel mucin domain proteins have been already reported in more recent data sets.

- Shon et al. 2020, PNAS indicated that BT4244 could bind to K562 after sialic acid treatment. Did the authors consider testing the enrichment strategy using BT4244 on K562 after sialidase, or on COSMC knockout K562 (+ sialidase) to confirm that BT4244 can similarly enrich glycoproteins?

- How does enrichment using an inactivated mucinase compare to enrichment using PNA or VVA? Does sialidase treatment affect the enrichment?

- Figure 3A-D, Are there mucin-like proteins that are not enriched reproducibly? Labels for these proteins would be useful as investigations as to why they are not enriched could be interesting.

Significance

- Using an inactivated mucinase to enrich for densely glycosylated proteins is a novel approach to this problem as compared to using lectin enrichment.
- The inactivated mucinases clearly specifically enrich for a subset of proteins, but what the exact property the mucinases enrich for is still unknown (beyond enrichment for more densely glycosylated proteins).
- This technique is interesting, and there are a number of applications for this (especially in the space of screening for changes to mucous environments in disease) that are possible. This research lays the foundations for further investigation into the area, and once the features that these mucinase enrich for are established (i.e. glycan type and density), the full relevance of these tools can be appreciated.

Summary of all major changes to the paper:

- New mucin-domain candidacy algorithm (Supp Data 1)
- Reanalysis of all data with the new algorithm; new figures which reflect this analysis (Fig. 2-5)
- Addition of Jacalin enrichment for comparison (Supp Fig 3, 4)
- Substantial analysis of enriched non-mucin proteins, overlapping mucin-domain proteins, and unenriched mucin-domain glycoproteins (Supp Data 4, 5, 7)
- Addition of sialidase pretreatment before BT4244 enrichment (Supp Fig 2, 4)
- Thorough investigation of glycopeptide enrichments (Fig 6D, E and Supp Data 8-12)

Reviewer #1 (Remarks to the Author):

In their manuscript „Revealing the human mucinome” Malaker et al. described an impressive approach to enrich mucin domain glycoproteins from cell cultures and ovarian cancer patient ascites fluids. The authors exploited the mucin domain binding capacity of a catalytically inactive mucinase followed by separation, in-gel digestion and LC-MS/MS analysis of the isolated protein samples. Comparison to untreated controls using label-free protein quantification showed the enrichment of 20-50 mucin domain glycoprotein candidates in the different samples. Furthermore, the authors developed an elaborate algorithm to predict mucin domains in glycoproteins and used this algorithm to corroborate the enrichment of potential mucins. Finally, the authors also identified glycopeptides to demonstrate the mucin-like glycosylation pattern of glycoproteins enriched from patient ascites fluid samples. Considering the relevance of mucins in disease pathologies and immune responses and currently limited knowledge on the mucinome this manuscript is timely and the novelties justify publication with minor modifications.

We thank the reviewer for their kind comments as well as their time and effort spent reviewing this paper.

Although it might seem a minor technicality, my major concern is the reliability of data interpretation used for the label free quantification. In the Methods sections the authors wrote that „precursor ion search tolerance of 20 ppm and a product ion mass tolerance of 0.3 Da were used for searches”. As the Orbitrap analyzer was used for MS2 data acquisition the mass errors should be well below 20 ppm. Why did the authors used this high product ion mass tolerance? 0.05 Da should have been still very permissive if relative mass accuracy could not be defined. If there was no second round of peptide identification with stricter parameters the reliability of the protein identifications is disputable. If this is the case the authors should reanalyze the data (or at least demonstrate on one particular dataset that the results are very similar with strict and loose search parameters). I did see that the authors applied FDR thresholds but the mass accuracies also affect the search space and hence the peptide FDR values.

Thank you to the reviewer for catching our mistake in reporting. We erroneously reported the ion trap MS/MS search settings in MaxQuant, but we also had an FTMS product ion tolerance setting of 20 ppm in the MaxQuant parameters. The program looks at scan headers to autonomously select the appropriate setting (either ion trap or FTMS), so we can assure the reviewer the product ion tolerance of 20 ppm was used to search the Orbitrap MS/MS scans, as they have suggested is appropriate. We have corrected this mistake in reporting in the methods section.

My minor questions and comments:

1. Canonical mucins are usually very difficult to digest due to heavy O-glycosylation of the mucin domains. Here, the authors could identify 59 O-glycopeptides from Mucin16. Were these peptides multiply glycosylated?

Unfortunately, because we used HCD for fragmentation, we are unable to definitively determine whether these peptides were multiply glycosylated. The complete list of glycan compositions is shown below:

Composition	Glycan Mass
N1	203
H1N1	365

N2	406
H1N2	568
N3	609
H1N1A1	656
H2N2	703
H1N2A1	859
H1N1A2	947
H2N2A1	1021
H2N2A2	1312

Additionally, the ratio of 138/144 in all of these cases is ~1, suggesting that the glycans are primarily core 1. Together, this would suggest that the compositions N2, H1N2, N3, H2N2, H1N2A1, H2N2A1, and H2N2A2 are multiply glycosylated peptides. This information has been added to the manuscript.

(Unfortunately, I could not access the relevant data in the repository. In fact, access to data in the Pride repository is very frustrating. One cannot search for specific files, max 40 files/page can be viewed at a time and any arrangement is possible on the page level. Please include the list of O-glycopeptide identifications as supplementary material along with all necessary data to enable manual assessment of glycopeptide Ids from raw data deposited.)

We have added supplementary data files (**Supplementary Data 9** and **10**) with N- and O-glycopeptide identifications, respectively, to make accessing this data easier. These files include identifications from the unenriched and enriched ascites samples; tabs for the overlap calculations discussed below and shown in newly made **Supplementary Figure 9**.

2. It would be interesting to see the mucin scores for the novel mucins described in the results. Please include these in the text and also in Suppl Tables 1-3.

We thank the reviewer for this suggestion. We have added this information in **Supplementary Data 7** as a replacement for Supplementary Tables 1 and 2. This dataset includes MucinScores for new (and known) mucins from cell line and ascites enrichments. Additionally, we have updated Supplementary Table 1 (formerly 3) to include MucinScores.

3. On p9 the authors describe their preliminary results using a mutant mucinase presumably binding mucin domain glycoproteins decorated only with truncated O-glycan structures (T and Tn antigens). The authors show that only a limited number of potential mucins are enriched significantly and claim that this is in good agreement with the expected non-truncated O-glycosylation pattern of HeLa glycoproteins. However, without demonstrating that the enriched mucins indeed display truncated glycans these results are somewhat unreliable. Furthermore, the portion of the enriched mucins in this particular sample (7/37, ~20%) is similar to the portion of depleted mucins from cell lines using the mutant SctE (30-50%), therefore it could be argued that the 7 enriched mucins are kind of „overall FDR mucins” similar to the mucins depleted in enriched samples using the SctE mutant. The findings described by the authors could be a starting point for another study aiming the isolation and site-specific characterization of truncated cancer-specific glycoforms, but does not add to the story of the current paper in my opinion, I suggest removing this part.

We thank the reviewer for their thoughtful comment on the enrichment using the inactive point mutant of BT4244. We further expanded our dataset with new data for inactive BT4244 enrichment with and without sialidase treatment prior to enrichment. Newly added **Supplementary Figs 2** and **4B** show that sialidase pretreatment did aid in the enrichment of mucin-domain glycoproteins, demonstrating to some degree the point we were originally trying to convey. Unfortunately, it appears as if BT4244 is less robust at enrichment of mucin-domain glycoproteins, especially when compared to StcE. However, we believe this serves as a proof-of-principle that other enzymes could potentially be used for the purpose of enrichment of mucins bearing truncated O-glycans. Thus, even though it is less robust than StcE, we believe it is still important to include conceptually. To address the reviewer’s comment further, we added the following to the manuscript:

“We then pre-treated HeLa lysate with 100 nM sialidase overnight and repeated this procedure, which resulted in the enrichment of 13 mucin-domain glycoproteins. Though not as robust as StcE enrichment, this proof-of-principle procedure demonstrates that other O-glycoproteases could be used to enrich and identify cancer-associated glycoforms of mucin-domain glycoproteins.”

4. Figure 6D is difficult to interpret. First, the same width for both N- and O-glycopeptide IDs visually suggests that the same number of N- and O-glycopeptide identifications were obtained. Second, it is impossible to assess the differences between the enriched/ascites samples. It is evident that N-glycopeptides are more prevalent in the ascites samples. On the other hand, it seems that there is a higher number of O-glycopeptide IDs in ascites compared to enriched sample in OC235. Please include total number of glycopeptide IDs for each sample.

We understand that interpretation of former Figure 6D was challenging, so we reimagined how to present the data. We now present **Figure 6D** and **6E** using visualization adapted from that have already proven useful in other glycoproteomics applications. These two panels show glycopeptide-glycan networks that illustrate which glycans map to which glycopeptide IDs. We have added the following figure legend and description in the main text to match:

Updated figure legend for Fig. 6D and E: Glycopeptide-glycan networks in panels **D** and **E** compare N- and O-glycopeptides, respectively, in ascites and enriched samples. For each glycopeptide-glycan network, unique glycopeptides (i.e., peptide sequence and total glycan mass) are organized vertically in the middle, and unique total glycan masses are nodes arranged in the outer semi-circles. The left and right semi-circles are mirror images of each other, showing the same glycan masses in the same order on either side, as indicated by numbers. The left side (gray) shows which glycan masses map to glycopeptides from ascites samples, and the right side (green or red) shows which glycan masses map to glycopeptides from enriched samples. These figures indicate which glycopeptides are shared or unique between the unenriched and enriched conditions and show that O-glycopeptides are detected more often than N-glycopeptides in mucin-enriched ascites fluid.

Supplementary Data 11 and **12** provide total glycan mass composition (outer nodes) identities and the unique glycopeptide list (middle nodes) for N-glycopeptide (**D**) and O-glycopeptide (**E**) networks, respectively.

Updated text:

“To visualize the degree of uniqueness/overlap between glycopeptides identified in ascites and enriched samples, we constructed glycopeptide-glycan networks shown in **Fig. 6D** and **6E**, which are modified versions of previous protein-glycan visualizations introduced in Riley et al. In these networks, unique glycopeptide identifications are arranged vertically as nodes in the middle of the network (black nodes in both panels). Unique glycan masses are then organized as nodes in the semi-circles on either side of the glycopeptide identifications, with each semi-circle representing the same glycan masses. In other words, gray nodes on the left of each network and color nodes on the right show the same glycan masses and are mirror images of each other. If glycan masses map to the same glycopeptide identifications, that means identifications are shared between the ascites (left, gray) and enriched (right, color) conditions. Otherwise, glycopeptide-glycan connections that only appear on one side of the network are unique to that condition. In **Fig. 6D**, the majority of N-glycopeptides were identified in ascites rather than enriched samples, with relatively few N-glycopeptides mapping uniquely to the enriched samples. Conversely, **Fig. 6E** shows that the majority of O-glycopeptides were identified in the enriched samples, with the majority of those being unique to the enriched samples. Note, **Fig. 6C** denotes glycoPSMs whereas **6D** and **6E** are unique glycopeptide identifications.”

5. Also, it would be interesting to see the overlap between O-glycopeptides (sequence+net glycan composition) identified from the different enriched ascites samples. Is there a set of peptides that were identified in all samples? (Figure 6D suggests significant biological variance.)

We added **Supplementary Fig 9** to describe the overlap in glycopeptide identifications. As noted above and in the manuscript, these are non-localized glycopeptides, so we are only considering base peptide sequence and total glycan mass when looking at overlapping identifications. We have also added **Supplementary Data 9** and **10** to supplement information provided by these graphs.

Supplementary Figure 9. Overlap in glycopeptides between ascites samples. Bar graphs show the number of N-glycopeptides (top, green) and O-glycopeptides (bottom, red) that were detected in n number of patient samples either in the unenriched ascites fluid (ascites) or the mucinome enriched samples (enriched).

“**Supplementary Fig. 9** (data available in **Supplementary Data 9** and **10**) shows the number of N- and O-glycopeptides detected in n number of experiments, suggesting a significant biological variance in glycopeptides between patients despite high protein-level overlap observed in **Fig. 5F**.”

Could these be used a biomarker panel to diagnose ovarian cancer?

Our current patient cohort is too small to suggest any clinical relevance. However, we are currently exploring this avenue by expanding the number of patient samples.

Are there any glycoproteomic data available for other pathologies resulting in ascites such as cirrhosis or other cancer types?

From what we have observed, all the relevant datasets available for comparison are for N-glycoproteomics, where O-glycopeptides and mucin-domain glycoproteins are notably underrepresented. Thus, we could not perform a straightforward comparison with published dataset to look at mucin-domain glycoproteins.

6. What was the overlap of enriched mucin domain glycoproteins and glycopeptides identified from the ascites samples (what portion of the O-glycopeptides represented the enriched mucin domain glycoproteins)?

We have included this data in **Supplementary Figure 9** and the new additional **Supplementary Data 9** and **10**. We have also added the following to the text:

“Slightly over 50% of all N-glycopeptide identifications in the enriched samples belonged to mucin-domain glycoproteins, while mucin-domain glycoproteins accounted for only ~15% of N-glycopeptide identifications from unenriched ascites fluid samples (**Supplementary Fig. 9**). For O-glycopeptides, approximately two-thirds (~66% and ~65%, respectively) of identifications from enriched and unenriched samples belonged to mucin-domain glycoproteins, but approximately 50% more O-glycopeptides could be identified from the enriched samples (**Supplementary Fig. 9**).”

7. What is the appr. protein concentration in ascites fluid and to what extent might it vary between patients? Was there any total protein amount estimation of the ascites fluid samples? If yes, please include.

The approximate protein concentration of the ascites fluid was 52 mg/mL, ranging from 33-64 mg/mL; this information has been added to the methods.

MS method description indicates that peptide-level BCA yielded highly varying total protein amount of ascites fluid samples.

Prior to performing peptide-level BCA analysis, we were using 6.5 uL, but this amount contained high levels of serum and resulted in column clogging. Thus, we added a peptide-level BCA step, which allowed us to inject approximately similar amounts (~1 µg) for each sample. We have added this information to the methods section.

8. Please include the O- and N-glycan databases considered for O-Pair searches in the Supplementary. How did the authors decided which 12 O-glycans to consider?

We have added the N- and O-glycan databases as **Supplementary Data 8**. We selected the 12 O-glycans based on previous literature, which were chosen due to common occurrence of these glycans in mucin domains. We have noted this in the methods section.

9. Mucin scoring indicates that there are some 350 human mucin domain glycoproteins. The current study identified ~50 of these. Could the authors comment on the „missing” 300? Are there any info on the expression level and/or trypsin accessibility of the „missing” proteins?

The missing 300 mucin-domain glycoproteins were likely not detected for a number of different reasons. First, we only explored 5 types of epithelial cancer cells; many other cancers and subsets of the same cancers are likely to express a different subset of mucin-domain glycoproteins. Also, we primarily used cell lysates in this study; given that a large portion of mucins are secreted, it is possible that we missed a large number of mucins only found in the secretome of cells. Related to this, it is widely accepted that immune cells express mucin-domain glycoproteins that are unlikely to be present on the cell types that we studied. Further, as the reviewer alludes to, it is entirely possible that the dense glycosylation in the mucin-domain glycoproteins renders them inaccessible to in-gel digestion using trypsin. Additionally, though previous experiments have suggested otherwise, it is possible that StcE is not a pan-mucinase and instead selectively enriches only a certain subset of mucins from the samples. Finally, as mentioned in the text, our mucin definition program is not perfect, and may be over (or under) estimating the total number of mucin-domain glycoproteins.

We have added an additional paragraph to the discussion to address these points:

“Though we have identified a subset of putative mucin-domain glycoproteins determined by the candidacy algorithm, we did not detect nearly 300 of these proteins. This can be likely be attributed to a number of reasons: first, we only explored 5 types of epithelial cancer cells; many other cancers and subsets of the same cancers are likely to express a different subset of mucin-domain glycoproteins. Also, we primarily used whole-cell lysates in this study, biasing toward membrane-tethered glycoproteins; given that mucin-domain glycoproteins can also exist as purely secreted biomolecules rather than membrane-tethered, it is possible that we missed a large number of mucin-domain glycoproteins only found in the secretome of cells. Further, it is entirely possible that the dense glycosylation in the mucin-domain glycoproteins renders them inaccessible to the in-gel digestion strategy used here. Current efforts are focused on optimizing the elution of the mucin-domain glycoproteins to enable in-solution digestion approaches. Additionally, though previous experiments have suggested otherwise, it is possible that StcE enriches only a certain subset of mucin-domain glycoproteins from the samples. Interestingly, during the review process of this manuscript, Nason, Büll, et al. reported that the C-terminal domain of StcE can confer mucin-binding properties irrespective of the active site⁶⁴, meaning that the selectivity of StcEE447D enrichments is not purely based on the O-glycosylated TxT motif that dictates its protease activity. This generates interesting new directions to explore complexities of mucin binding harbored by catalytically inactive O-glycoprotease mutants.”

Minor comments:

1. In Figure 2 caption and on page 7 the authors say „approximately 330 proteins contain a mucin domain by our estimate (score > 1)” – in suppl dataset 1 there are exactly 350 proteins with mucin score >1. Please correct.

We thank the reviewer for noticing this error and have made this correction to the number of 357 that has resulted from changes made during this review.

2. Page 8: „Several of the proteins were previously known to have a mucin domain”... as the authors discuss 6 proteins altogether, „several” does not seem to be the appropriate wording here. Please rephrase.

We apologize for the misunderstanding; we have edited the text to be clearer:

Several of the proteins (4/7) were previously known to have a mucin domain, including: Mucin-1 (MUC1), dystroglycan (DAG1), agrin (AGN), and complement decay factor (CD55, DAF). However, we discovered that three of the overlapping proteins have previously undescribed mucin domains: low-density lipoprotein receptor 8 (LRP8), major facilitator superfamily domain 6 (MSFD6), and porimin (PORIM).

3. In suppl dataset 2, „Capan2” tab need to be revisited. In the „IsaMucin” column many proteins are not classified, one is indicated as „UD14_HUMAN”.

Thank you for pointing this out. This has been corrected.

4. Please elaborate on „mucin score” values. How is „NaN” and „0” different?

NaN indicates a Mucin Score was not able to be calculated for a given protein because output was not retrievable from NetOGlyc. Of the 20,365 entries in the Uniprot human proteome (reviewed), 174 were unable to return a score from NetOGlyc, meaning they can be identified in our proteomics searches but will not have a mucin score associated with them (NaN). A score of zero indicates that NetOGlyc was able to return an output, but that protein returned a score of 0 based on the candidacy algorithm. We have added a sentence in the methods to explain this:

“NetOglyc4.0 results were saved as .csv files for further processing, with 20,121 entries returning usable output. Those without a NetOGlyc4.0 output received a Mucin Score of NaN in the supplemental datafiles, which differs from a score of 0 that can be calculated through the description below.”

5. In the cell line experiments, did the authors check for FBS contamination in the samples? In the methods section it is described that only the Uniprot/human database was considered in the database searches.

We did not check for FBS contamination. We have included this qualifier in the methods section.

6. Results section, p4: „Through a straightforward reductive amidation approach” – the protein is immobilized via an amine linkage, so this is a reductive amination not amidation, please correct.

Thank you for pointing out this error; we have corrected this.

7. Methods section: „Ascites fluid was obtained from O.D. and V.K. – please elaborate (patients? hospital administrators? etc). „Samples were selected based on their grouping in ref 28” again, please clarify.

O.D. and V.K. are authors on this paper and members of the Stanford Division of Gynecologic Oncology. Given the confusion, we have removed the statement about “grouping in Ref 28”.

8. Methods section, in-gel digestion: it is written that following tryptic digestion and acidification the supernatant of the gels were discarded and only the ACN extract was kept for downstream LC-MS/MS. In my experience, the aqueous „extract” contains a large portion of the tryptic peptides, therefore it would be interesting to know why these samples were excluded from analysis. Please clarify.

The protocol we developed our in-gel digestion from did not save these fractions. These fractions will be kept for future experiments, and we appreciate the reviewer making us aware of this.

9. LC-MS: solvent B, 2% formic acid in acetonitrile – is that correct?

Thank you for pointing out this oversight. We have corrected this to 0.1% formic acid in acetonitrile, which was the buffer actually used.

10. LC-MS: „MS1 resolution of 60,000 at FWHM 400 m/z” – the resolution is defined at m/z 200 for the instrument used in the experiments.

Thank you for pointing this out, we have made this correction.

11. It is a bit difficult to interpret data included in the Suppl datasets. First column: „significant”, second column: (-LOG(P-value)), third column: Difference - what these columns mean? How are these values calculated, what cut-offs were used? Please include legends to explain data in the Supplementary dataset tables.

Thank you for helping us to clarify this reporting. We have added this to the methods section to explain these headings which appear in multiple supplemental data files:

“We kept the standard Perseus column headers from these analyses, with “Significant” showing a “+” for proteins calculated as significant based on the t-test performed, -log(P-value) providing the y-axis value of the volcano plots that shows the log-transformed value of the t-test p-value, and “Difference” indicating the log₂ fold change between the conditions (e.g., elute and lysate).”

Reviewer #2 (Remarks to the Author):

The Bertozzi group has used inactive StcE lectin to purify and analyze proteins and tryptic peptides from cell lines and ascites. By this they characterized a number of proteins in a few cell lines and in ovarian cancer ascites previously not known to be O-glycosylated as listed in the Simple Cell dataset. They have analyzed the samples by mass spectrometry and the hits by various criteria and bioinformatics.

We thank the reviewer for their time and efforts spent reviewing this paper.

This is a potentially interesting publication, but also contain some mistakes and over estimations. To use lectins to purify glycoproteins has been used for many years where some plant lectins, like Jacalin, was thought to be more specific for mucin domains and thus the approach is not new.

We respectfully disagree with the reviewer that the approach is not new. Jacalin is a lectin that will enrich for the T and Tn antigen (i.e. GalNAc-Gal, GalNAc) **glycans** but is not specific for mucin **domains**. Jacalin is likely biased toward mucin domains due to the high number of these modifications on these proteins, but definitely enriches non-mucin domain proteins. Further, the novelty relies in this case on enriching for a functional domain, as opposed to a subset of glycans.

To demonstrate this point, we conjugated Jacalin to POROS-AL beads and performed an enrichment on HeLa lysate +/- sialidase pre-treatment. The enrichment was performed exactly as the StcE pulldown. The volcano plots for these enrichments are shown in **Supplementary Figure 3**. To be sure, Jacalin does enrich most of the mucin-domain glycoproteins, but as demonstrated by the large number of enriched non-mucin proteins, it is clear that Jacalin is less specific for mucin-domain glycoproteins. This point is further illustrated in **Supplementary Figure 4**. The Jacalin (+/-) pulldown resulted in the enrichment of 205 and 273 proteins, respectively. The percentage of mucin-domain glycoproteins within this subset is only 16-17%, meaning that 171 and 230 non-mucin proteins were found in the two samples.

Using the same HeLa lysate, StcE-conjugated beads enriched a total of 75 proteins, 28% of which were mucin-domain glycoproteins. Thus, StcE roughly twice as selective for mucin-domains. Further, we detected only 54 non-mucin proteins in this enrichment, compared to the 230 in the Jacalin pulldown. This is an over 4-fold reduction in non-mucin proteins. Given that our ultimate goal is to identify glycopeptides from the mucin-domain glycoproteins, selectivity is extremely important. Non-mucin proteins, and their associated unmodified peptides, will outcompete the glycopeptides for ionization and detection.

This language has been added to the manuscript and accompanying Supplementary Figures have been added to the manuscript:

“We next asked how selective our mucinomics platform is when compared to lectin enrichments commonly used for O-glycoproteomics. Jacalin has preference for proteins bearing mucin-type O-glycans including GalNAc and GalNAc-Gal; thus, we conjugated Jacalin to POROS-AL beads and performed enrichments on HeLa cell lysate with and without pretreatment with sialidase. The resulting volcano plots are shown in **Supplementary Fig. S3**. To be sure, Jacalin does enrich most of the mucin-domain glycoproteins, but as demonstrated by the large number of enriched non-mucin proteins, it is clear that Jacalin is less specific for mucin-domain glycoproteins. This point is further illustrated in **Supplementary Fig. S4**. The Jacalin (+/- sialidase) pulldown resulted in the enrichment of 205 and 273 proteins, respectively. The percentage of mucin-domain glycoproteins within this subset is only 16-17%, meaning that 171 and 230 non-mucin proteins were found in the two samples. Using the same HeLa lysate, StcE^{E447D}-conjugated beads enriched a total of 75 proteins, 28% of which were mucin-domain glycoproteins. Thus, StcE^{E447D} is more selective for mucin-domain glycoproteins. Further, we detected only 54 non-mucin proteins in this enrichment, compared to the 230 in the Jacalin pulldown, representing a >4-fold reduction in non-mucin proteins. This selectivity is especially important when considering potential goals of characterizing mucin-domain O-glycopeptides. Non-mucin proteins, and their associated unmodified peptides, will outcompete the glycopeptides for ionization and detection.”

Supplementary Figure 3. Jacalin enrichment of HeLa lysate. Jacalin was conjugated to POROS-AL beads using reductive amidation and HeLa lysate (A) or HeLa lysate pretreated with 100 nM VC sialidase overnight (B) was added to the beads. After binding, washing, and eluting, in-gel digest was performed on lysate alone and the elution of the enrichment. The samples were run on an Orbitrap Fusion Tribrid followed by a MaxQuant search. The data was processed using Perseus, and mucins were labeled according to the MucinScore. Red signified a score of >2 (high confidence), orange 2-1.5 (medium confidence), and yellow 1.5-1.2 (low confidence). Strongly enriched proteins are labeled with their gene names.

Supplementary Figure 4. Selectivity for mucin-domain glycoproteins in StcEE447D, BT4244E575A, and Jacalin enrichments. Statistically significantly enriched proteins were considered mucin-domain glycoproteins (red) if the MucinScore was higher than 1.2. All other proteins are considered non-mucin proteins (gray). The percentage of mucin-domain glycoproteins compared to total proteins is indicated on each bar. (A) StcEE447D cell line enrichments. The average selectivity for mucin-domain glycoproteins was 25.1%. More information regarding enriched proteins can be found in Supplementary Data 4, including mucin and non-mucin protein

identifications, MucinScores, and GO terms. (B) BT4244E575A and Jacalin enrichments. BT4244 was much more selective for mucins after sialidase treatment. Jacalin exhibited poor selectivity in both cases.

Although StcE clearly binds clustered O-glycans as found in mucin domains, it is highly unlikely that it will bind all such domains as binding will depend on the glycan structures and their arrangements. Thus we need to have better knowledge of the specificity of the StcE binding to know which mucin domains that are bound. The specificity of StcE is good and the approach is good, but the text is claiming that all mucin domains will be enriched is likely not the case. However, the authors have isolated a subset of mucin-domain containing proteins of which some were not described as having a mucin domain before.

In concordance with reviewer 1 above, we have added a paragraph in the discussion addressing the fact that we have not identified all 350 predicted mucin-domain glycoproteins. One of these reasons is likely due to StcE's bias toward specific subsets of the mucinome.

The purified proteins were separated on SDS-PAGE and all components cut out and analyzed by mass spectrometry. The enriched proteins were identified after trypsin cleavage by sequences outside of the mucin domains as these will be lost during the preparation and not possible to identify within the mass range studied. This has to be explained.

The interaction between StcE and its mucin-domain glycoprotein targets is incredibly strong, thus necessitating boiling in SDS to elute the proteins. Given that SDS is not a MS-friendly detergent, this then follows that an in-gel digest is necessary after the enrichment. Additionally, we have attempted to perform in-gel digest with StcE, StcE with trypsin, and trypsin alone. In this comparison, we found that the highest number of peptides generated was always in the tryptic samples, suggesting that StcE is not the best protease for in-gel digestion. We are aware that this sample processing format limits access to mucin-domain glycopeptides themselves, but it still enables specific enrichment and detection of mucin-domain glycoproteins from complex mixtures. We have added text to summarize this:

“Additionally, given that we performed in-gel tryptic digestion, it is unlikely that we were able to extract the intact mucin domains from many of our samples, nor were we able to fully characterize mucin domains of interest. Attempts to use StcE for in-gel digests resulted in limited digestion efficiency, and alternative methods to couple StcE proteolysis are currently under investigation.”

Most important, the authors have to present the mucin domain peptide sequences of the identified proteins allowing the reader to interpret their predictions of mucin domains.

As discussed above, due to the usage of trypsin, most of the peptides we are detecting were either mucin-domains that contained some basic residues interspersed or from non-mucin portions of mucin-domain glycoproteins. Our new supplemental data includes all the mucin candidate proteins and predicted sites in addition to the O-glycopeptide identifications (**Supplementary Data 10**), providing context for where our identifications were derived. Finally, as we note above, we were aware of this shortcoming and focused more on the protein-level identifications as opposed to the glycopeptide mucin domain.

This also means that the identification is not very different from other ways of mining DNA/protein sequences. Typical for mucin domain sequences are their lack of sequence conservation over evolution, long sequences, only proteins with signal sequence or transmembrane domains, and the high abundance of Ser and Thr amino acids interrupted by Pro. This rendered the name 'PTS sequences' that is preferred to the 'VNTR sequence' as many mucin domains lack an identifiable repeat structure. The characteristics for a PTS has been used for mining genomic databases by the Samuelsson group (Lang, T., Alexandersson, M., Hansson, G. C., and Samuelsson, T. Bioinformatic identification of polymerizing and transmembrane mucins in the puffer fish *Fugu rubripes*. *Glycobiology* 14, 521-527. 2004; Lang, T., Hansson, G. C., and Samuelsson, T. Gel-forming mucins appeared early in metazoan evolution. *Proc.Natl.Acad.Sci.USA* 104, 16209-16214. 2007). This is not discussed despite that the criteria for a mucin domain length and Ser/Thr content is similar to that used before.

We have added the following to the manuscript to make sure to highlight this important work while also emphasizing that our approach incorporated new features of predicted O-glycosites and protein-level information about both subcellular location and known phosphosites:

“Previous work has mined sequences looking for PTS domains in various non-human organisms, but we wanted to extend our criteria to use protein-level data that includes predicted O-glycosites, subcellular localization information, and previously annotated PTM-sites to annotate putative mucin domains in the human proteome.”

They relate their information to the Simple Cell database but this is partly irrelevant (Tables S1-S2) as this approach is excellent for identifying single O-glycans, but poor for larger clustered O-glycans as in mucin domains. The NetOGlyc4 tool also has limitations as this is not taking into account that there are 20 different peptidyl-GalNAc transferases with different specificities as pointed out by Gerken et al (Daniel, Earnest James Paul, Las Rivas, Matilde, Lira-Navarrete, Erandi, Garcia-Garcia, Ana, Hurtado-Guerrero, Ramon, Clausen, Henrik, and Gerken, Thomas A. Ser and Thr Acceptor Preferences of the GalNAc-Ts Vary Among Isoenzymes to Modulate Mucin Type O-Glycosylation. *Glycobiology* . 4-18-2020)

We respectfully disagree with the reviewer that the SimpleCell dataset is irrelevant for our study. The SimpleCell dataset identified many O-glycopeptides from mucin-domain glycoproteins across a number of different cell lines, allowing a comparison to our multiple cell lines; some of these glycopeptides were found within our assigned mucin domains. Additionally, the sites were confirmed using electron-based fragmentation, unlike several other O-glycoproteomic datasets available. Thus, we believe that the SimpleCell dataset is highly relevant for our work.

Another major issue for the manuscript is nomenclature. There is a set of human glycoproteins whose genes has been named mucins. This is a heterogeneous group of glycoproteins, but it does not help that the authors starts to call also additional proteins mucins. Stick with the nomenclature of ‘mucin domain containing proteins’ also for the figure 2 and text.

This is a good point. We agree that these additional proteins should be referred to as mucin-domain glycoproteins, and we have edited the manuscript to reflect this.

The authors define what a mucin domain is by stating that there should be >50 amino acid residues with predicted 10 O-glycan sites. All mucin domains so far are found in a single exon, something that should be analyzed for the new mucin domains.

We thank the reviewer for this suggestion. We manually checked putative mucin domains against known exons using the neXtProt knowledgebase, where we found 90.2% of our mucin candidate regions to come from one exon. Of the small fraction that did not come from one exon were just one or two amino acids different, showing a potential mucin domain that may have been slightly miscalculated based on NetOGlyc prediction. We have added the following text to the manuscript to address this comment:

“For regions of 50 amino acids identified as putative mucin domains through this analysis, approximately 90% of them mapped to a single exon, which was evaluated manually using the neXtProt knowledgebase⁶⁸.”

The use of the word ‘mucinome’ is acceptable, but mucinases is imprecise as it is often used for bacterial O-glycopeptidases and O-glycohydrolases. How about ‘mucin-domain peptidase’?

We have replaced mucinases with O-glycoproteases, which is often used to describe this family of enzymes.

I should avoid mucinopathologies, what is this?

Mucinopathies are diseases in which mucins are known to be dysregulated. We have added this definition to the text: “Further, while we chose to focus our efforts on the cancer mucinome, several other disease mucinomes have yet to be studied in diseases known to involve dysregulated mucins. These mucinopathies include, but are not limited to, inflammatory bowel disease...”

Mucinomics is also imprecise and suggest a method for identifying all mucin domains, something that is not possible today where we know that some mucin domains have molecular masses in the MDa range.

We respectfully disagree with the reviewer on the definition of mucinomics. Proteomics is a broad term to define the study of proteins in a given sample and does not necessarily identify every protein. Similarly, glycoproteomics is a term used to describe the glycoprotein analysis of samples but absolutely does not identify every glycopeptide in a given sample. Thus, we have not changed this terminology.

Page 3

- The abstract claim to identify mucins, should be removed.

We have changed all “mucins” to “mucin-domain glycoproteins” where appropriate.

- Not all mucin domains occur as tandem repeats – most of the mucin domains described here do not have tandem repeats.

In this sentence we are specifically referring to the canonical family of mucins, which are often found as tandem repeats. In order to clarify, we have added “often” to the sentence.

Page 4

- StcE does not cleave all mucin domains and is thus not a pan-mucinase.

We have removed all instances of “pan-mucinase” from the document.

Page 4

- Figure 1: Most mucin domain glycoproteins have their mucin domain close to the membrane to generate a stalk, like the LDI receptor

Thank you for this observation. We have changed Figure 1 to reflect this suggestion.

- The authors have not developed a mucin candidacy algorithm, maybe a mucin domain candidacy algorithm

We have changed “mucin candidacy” to “mucin-domain candidacy” throughout the manuscript.

- It is unclear why the authors chose to analyze for phosphosites as transmembrane proteins might have intracellular phosphorylation and extracellular O-GalNAc. Should it not be enough to analyze for secreted or transmembrane proteins that is done in next step?

In earlier iterations of the mucin-domain candidacy algorithm, we found that despite only looking at transmembrane and secreted proteins, NetOGlyc falsely assigned intracellular phosphosites as being O-glycosylated. When manually curating the outputs from the candidacy algorithm, we found that many membrane-associated proteins were actually proteins with multiple subcellular locations and large intracellular phosphosite domains, which were being mislabeled as mucin-domain glycoproteins. Thus, removing sites that have been curated as phosphorylated also removed these false positive hits from our list.

Page 5

- Mucin domains also has to contain a substantial frequency of Pro, why was this not considered?

We elected to use glycosite features of mucin domains as the defining characteristic around which to build our candidacy algorithm. Not only is dense O-glycosylation at serines and threonines the most widely recognized feature of mucins, but also this is a characteristic that could be aided by the NetOGlyc prediction tool. We have added language to the discussion to note this decision and that other features of mucin domains, such as the frequency of prolines, could be included in future mucin domain candidacy algorithms:

“We chose to build this candidacy algorithm on the hallmark mucin domain feature of serine and threonine O-glycosylation, as predicted by NetOGlyc4.0, while not focusing on other sequence characteristics such as proline frequency.”

- Fig. 2 Mucin should be replaced by mucin domain

We have changed all “mucins” to “mucin-domain glycoproteins” where appropriate.

Page 8

- Remove ‘pan-mucinase’

We have removed all instances of “pan-mucinase” from the document.

- Figure 4 is difficult to understand. Why the left hand subfigure? The presence of which of the 6 proteins shown are present in each of the 1-5 bars?

Upset plots are an increasingly popular way to compare different sets of data. The left-hand subfigure is to help the reader understand how to read the Upset plot; this is also described in the text:

The total number of enriched mucin-domain glycoproteins from each cell line is shown on the bottom left (blue horizontal bars). If a group of mucin-domain glycoproteins was only seen in one cell line, only one gray dot is darkened; the number of proteins that are only seen in that cell line are shown in bar graph form above. For instance, 6 mucin-domain glycoproteins were only detected in the K562 cell line, whereas 2 mucin-domain glycoproteins were only detected in both the SKBR3 and OVCAR3 cell lines. Overlap between samples are shown by multiple darkened gray dots and a line connecting them. **A total of seven mucin-domain glycoproteins were seen in all five cell lines; these proteins are shown above the Upset plot.**

Why are there 3 bars for the 6 proteins??

We are unclear as to what the reviewer is referring to in this question. Hopefully, the Upset plot description above helped to clear up any confusion.

Replace Mucins with mucin containing glycoproteins.

The figure has been edited as requested.

- The approach is mucin domain enrichment approach and nothing else.

We are unclear as to what the reviewer is referring to in this comment, but we believe we have made the changes as part of this review to make it clear that our approach is a mucin domain enrichment.

Page 10

- The tryptic fragments analyzed will not reveal the large classical mucin domains as these are too large and lost during preparation. Should be discussed instead of the ETD vs. HCD.

We believe that the HCD vs ETD discussion is important to address, so we have not removed that statement. However, we did add an additional sentence recognizing the additional drawback of our approach:

“Additionally, given that we performed in-gel tryptic digestion, it is unlikely that we were able to extract the intact mucin domains from many of our samples.”

Page 11

- Fig. 6. Replace mucin. The word mucin should be replaced by mucin domain in almost all instances throughout the text.

We have changed all “mucins” to “mucin-domain glycoproteins” where appropriate.

Reviewer #3 (Remarks to the Author):

The manuscript from Malaker & Riley et al. is a well written manuscript that uses a novel approach to develop a tool that can be used to reveal an under-appreciated part of the proteome: the “mucinome”. I appreciate the general approach to this problem, but I believe that there are a few points that need to be clarified in this manuscript so that the full potential of this work can be realised. If it was possible to analyse the data to give more indications as to what the exact patterns are that the inactivated mucinases/glycopeptidases are enriching for, I think this would improve this manuscript. For example, do they enrich for a particular O-linked glycan on the proteins, are sequence motifs (e.g. TxT) recognised by the inactivated mucinases, and so enriched for in the resultant proteins?

The authors have decided upon the presence of mucin domains (defined essentially as a minimal density and length of glycosylated regions) as the major factor that contributes to this enrichment. I would like to see a bit more information about alternative hypotheses (all examined bioinformatically) to explain the enrichment, especially taking into account what is known about the specificities of the active mucinases. While the data as it stands may not allow for insight into which elaborated/truncated glycans are present, information about potential density of glycosylation and motifs could be extracted, and compared to the mucin-domain candidacy algorithm.

We thank the reviewer for their kind comments as well as their time and effort spent reviewing this paper.

Key results

- The mucinome is a subset of the human proteome that is relatively poorly characterised, and can be revealed using mass spectrometry and selective enrichment using inactivated mucinases
- The mucinome can be enriched for using catalytically inactivated mucinases (StcE & BT4244).
- They enrich mucins because there are more proteins classified as mucin/mucin-like in the enriched portions.
- The classification of mucin and mucin-like domain containing proteins uses a mucin-domain candidacy algorithm.
- The enrichment can define key molecular signatures of cancer in-vitro / ex vivo.

Nomenclature / naming:

• Nomenclature is always a difficult discussion to have because the reasoning behind preferences is often highly subjective (highly dependent on experience and research focus). However, I think the usage of “mucin” as opposed to “mucin-like” to describe regions of dense O-GalNAc glycosylation has the potential to introduce confusion. Mucins have conventionally been defined as proteins that contain tandem-repeats with a high density of PTS, and the repeating region is integral to protein identity. Complicating this, there are proteins called mucins, that don't have repeats (e.g. MUC13), or proteins that have PTS repeats but aren't mucins (e.g. GP1Ba or CSPG). Furthermore, a number of proteins have “mucin-like” domains (e.g. DAG1 and KIAA1549), characterised by large amounts of O-glycosylation (in both these cases, they carry O-Mannose and O-GalNAc residues). Finally, many proteins have shorter regions that have high densities of glycosylation, often found as linker regions between domains, stems on Type 1 or 2 proteins or on N- and C- termini of proteins (DOI: 10.1007/978-4-431-54841-6_63). All these can have high density of glycosylation in longer stretches, but they're not all mucins.

To obviate some of this confusion and also address Reviewer 2's comments, we have changed all instances of “mucins” to “mucin-domain glycoproteins”, unless specifically referring to the canonical family of mucins.

Functionally, the mucins fulfil specific roles within their respective niches, with function derived from not only the high density of glycosylation, but likely also impacted by the tandem repeats on the protein. Overloading the meaning of the mucin to include proteins that are likely much more functionally diverse will conflate the functions that are commonly associated with mucins with any region that has a high density of glycosylation. The term

“mucin-like” does not have this problem as it has been historically used to describe regions carrying large amounts of O-GalNAc glycosylation, but is an awkward term to use as an ‘ome.

We address this point, albeit indirectly, in the discussion. Our thought regarding the “mucinome” is that select subsets of mucin domains (within the mucin-domain glycoproteins) are likely to function similarly within their environment. Given that mucin domains, especially within the large MDa canonical mucins, are enigmatic to study, it is useful to know all of the proteins that contain a mucin domain so that the field may begin to pare down the number of potential actions that mucins play. In this sense, the mucinome connects all proteins containing a mucin domain (i.e., mucin-domain glycoproteins) that can be further parsed into functional roles, such as biophysical drivers or biochemically active mucins – not unlike how we generally describe other “-omes” (e.g., the glycoproteome) that can be delineated further into subsets.

If the authors have other suggestions for a specific umbrella term to describe these regions of high density glycosylation (which could be further sub-divided in to mucin, mucin-like and short mucin-like) this would sidestep these issues.

We hope that by replacing mucins with mucin-domain glycoproteins, we can address the reviewer’s perspective on this and use a more precise term that captures the information we hope to convey.

Validation of enrichment:

- The inactivated mucinases enrich a subset proteins, but it is not known whether they specifically enrich for mucins or mucin-like proteins.

In Figure 2A, we demonstrate that all of the canonical mucins present in the HeLa lysate are specifically enriched. Then, we go on to show that other mucin-domain glycoproteins are also enriched. This observation holds true for other enrichments (lysates and ascites fluid) throughout the manuscript. Thus, inactive StcE can enrich for both mucins and mucin-domain glycoproteins alike.

The authors suggest that the presence of mucin domains on proteins lead to the enrichment and thus categorise proteins into groups depending on the likelihood that they have mucin domains. The analysis assumes that this categorisation system that they use is the most appropriate, but any comparison of using this categorisation, as compared to other potentially simpler systems is not provided. There are a few simple categorisations that could be relevant to compare the mucin-domain candidacy algorithm against.

From the previous literature (Malaker et al. 2019, PNAS, Shon et al. 2020, PNAS), the specificity of the mucinases. StcE is primarily towards a TxT motif (with glycosylation), while BT4244 prefers a glycosylated threonine.

While true, the specificities listed here are the active site cleavage motifs; that is, where along a peptide backbone StcE and BT4244 will cleave. However, given that StcE and BT4244 have been crystallized, we know that the active site is not solely responsible for the selectivity toward mucin domains. For instance, StcE has a C-terminal domain that is known to confer binding to mucins; in fact, when expressed alone, authors have demonstrated that this domain will selectively bind to mucins (Nason, R., Büll, C. *et al.*, Display of the Human Mucinome with Defined O-Glycans by Gene Engineered Cells. 2021, 12: 4070, *Nat Comm.*). Similarly, BT4244 harbors two IG domains that are likely to be carbohydrate binding domains. Other O-glycoproteases, such as the Zmp family from *C. perfringens*, harbor many carbohydrate binding domains that can direct the mucin to the active site for cleavage (Pluvinage *et al.* Architecturally complex O-glycopeptidases are customized for mucin recognition and hydrolysis. 2021, PNAS). Taken together, the mucin selectivity of these enzymes is not solely the active site recognition sequence, but instead relies on the larger interaction of the enzyme structure with the mucin domain(s). We have added language to address these ideas to the Discussion.

Two metrics could be derived: the counts (abundance) of TxT motifs, and the counts of TxT motifs that are predicted to be glycosylated (using e.g. NetOGlyc). Are there any correlations between the number of TxT motifs and the resultant quantitation of protein enrichment?

We thank the reviewer for their thoughts here, but as mentioned above, we have come to appreciate that the active site is not the sole contribution for binding. The C-terminal domain that is not involved with recognition of the TxT motif for cleavage also confers mucin binding properties and is indeed enough for selective mucin binding. Because of this, we think that calculating the counts of TxT motifs as the reviewer suggested would only produce confounding data that would not fully explain binding observations. For that reason, we have not added these calculations to the manuscript.

Another metric that could be used is the total relative / absolute length of regions of high density of glycosylation on proteins. If you look at MUC4 and MUC16, you can expect that approximately 6/10 amino acids are possibly glycosylated in their repeat regions (and 5/10 amino acids for MUC1). If you run a sliding window across the protein sequence, you can calculate these densities for proteins. Similarly to the mucin candidacy algorithm suggested in the manuscript, these calculations can be limited to extracellular regions.

We thank the reviewer for this point. This appears to be a similar idea to how we arrived at our calculations of the 9 O-glycosites within a sliding window of 50 residues. Within the MUC4 and MUC16 examples, this would account for ~30 O-glycosites within any given 50 residue window in the repeat regions, which clearly marks those proteins as highly probable mucins in the algorithm. Furthermore, as the reviewer suggests, we require an O-glycosite density within these regions so that there is not a gap of more than 12% (or 6 residues in a 50 residue stretch) between any O-glycosites, which we believe also captures the reviewer's ideas of considering densities.

- The authors do not comment on how many (or what proportion of) proteins in the enriched portion do not have mucin-like properties. Quantification of this will provide for a confidence that the inactivated mucinases specifically enrich for mucin/mucin-like domains.

We thank the reviewer for making this important point. To address this suggestion, we have investigated the quantity of non-mucins enriched by StcE. This information can be found in Supplementary Figs 4A (cell lines) and 7 (ascites). In the cell lines, the average percentage of mucin-domain glycoproteins compared to non-mucins was 25%, ranging from 22-29%. To compare the selectivity of our platform to commonly employed enrichment techniques, we performed a Jacalin enrichment (Supplementary Figs 3 and 4B). In the Jacalin HeLa enrichments, over 200 non-mucin proteins were identified versus approximately 50 in the StcE enrichment. Thus, StcE is far more selective than Jacalin in enriching mucin-domain glycoproteins from HeLa cell lysate. Further, the enrichment is even more selective in less complex samples such as cancer patient ascites fluid, where the average selectivity was 43.4%. Overall, we believe this demonstrates the selectivity for mucin domains.

- For those non-mucin proteins that are reproducibly enriched, and lack a TxT motif, do the authors have any theories as to why these proteins are enriched? Is this an indication of a broader specificity of the enrichment behaviour of the inactivated mucinase? I have taken a brief look at the supplemental tables, and it looks a lot like there are many nucleocytoplasmic proteins that are enriched. If this is the case, then maybe these proteins could be recoloured/de-emphasised in the volcano plots as they are unlikely to be directly enriched by the mucinases? (Unless somehow the mucinases enrich for O-GlcNAc too).

StcE is selective for extracellular mucin-type glycosylation and is not effective on O-GlcNAc or O-mannose modified peptides, as demonstrated in Malaker, *et al.* Proc. Natl. Acad. Sci. U. S. A. 116, 7278–7287 (2019).

To investigate the reviewer's question, we thoroughly investigated the non-mucin content of the enrichments; this information is now contained in **Supplementary Figs. 4-8** and **Supplementary Data 4** and **7**. We first investigated how many of the "non-mucins" were commonly found between cell lines, as demonstrated by the Upset Plot in Supplementary Fig. S5. Here, the majority of proteins were found in only one cell line, suggesting that these proteins are primarily non-specifically binding to the beads. On the other hand, 5 proteins were found in all cell lines, and 7 were found in at least 4 cell lines (**Supplementary Data 4**; Master_NonMucin tab). Of these 12 proteins, 6 are likely to be underscored mucin-domain glycoproteins. The other proteins are likely to be (a) abundantly expressed and non-specifically binding (e.g. myosin) and/or (b) previously undescribed glycan or mucin-binding proteins. Taking this one step further, we performed DAVID GO_CC term enrichments for all of the non-mucins. The highest protein counts were "extracellular exosome" (87) and "integral component of membrane" (80); "perinuclear region of cytoplasm" is far less abundant at a protein count of 15.

Similarly, in the ascites samples, the majority of enriched non-mucins were found in only one sample; however, here 13 proteins were found in all 5 samples and 11 in at least 4 (**Supplementary Fig. 8, Supplementary Data 7; Master_NonMucin tab**). Of these, we can attribute six proteins as either underscored mucins, lectins, or glycan/mucin-binding proteins. The rest are largely extracellular or transmembrane proteins; this is also demonstrated by the enriched GO terms. The most enriched terms are “extracellular exosome” (100), “extracellular region” (57), and “extracellular space” (52).

Supplementary Figure 5. Upset plot comparing enriched non-mucin proteins from five cell lysates. Figure was generated using Intervene UpSet tool. The majority of non-mucin proteins were found only in one cell line; however, 5 non-mucin proteins overlapped in all five lysates. Information about the overlapping non-mucin proteins can be found in Supplementary Data 4, including Uniprot IDs, protein names, and enriched GO terms.

**Supplementary Figure 6. Upset plot comparing unenriched mucin-domain glycoproteins from five cell lysates.** Figure was generated using Intervene UpSet tool. Five mucin-domain glycoproteins were consistently not enriched in the cell lysate samples. Information about these five proteins can be found in Supplementary Data 5, including Uniprot IDs, protein names, and enriched GO terms.

**Supplementary Figure 7. Selectivity for mucin-domain glycoproteins in StcE$^{E447D}$ ascites enrichments**. Statistically significantly enriched proteins were considered mucin-domain glycoproteins (red) if the MucinScore was higher than 1.2. All other proteins are considered non-mucin proteins (gray). The percentage of mucin-domain glycoproteins compared to total proteins is indicated on each bar. The average selectivity was 43.4%.

Supplementary Figure 8. Upset plot comparing enriched non-mucin proteins from five ascites patient samples. Figure was generated using Intervene UpSet tool. The majority of proteins are found only in one sample; however, 13 non-mucin proteins overlapped in all five patient samples. Information about the overlapping non-mucin proteins can be found in Supplementary Data 7, including Uniprot IDs, protein names, and enriched GO terms.

- Figures 2A and C could be reworked so that it is easier to benchmark the performance of the mucin candidacy algorithm. It would be useful to be able to evaluate whether the algorithm can successfully identify the known enriched mucins (these points remain green in Figure 2C, so you cannot evaluate the confidence that these proteins are mucins).

All of the green proteins are high confidence mucins.

Secondly, how many proteins are both enriched, and not identified as mucin-domain proteins? In Figure 2A, clearly there are many more proteins not identified as mucin than are mucin, but it is harder to derive this information from Figure 2C. If both the mucin-domain candidacy algorithm works, and the inactivated StcE enriches mucin/mucin-like domain containing proteins, I would expect that there are more proteins predicted to contain mucin-like domains in the enriched portion than proteins not predicted to be mucins.

We respectfully believe that the reviewer has misunderstood Figure 2. For instance, Figure 2A and 2C are the exact same volcano plot; thus, the same number of mucins are enriched in both A and C. The difference is that prior to the development of the mucin-domain candidacy algorithm, we were at the mercy of those proteins that were previously annotated as mucins (indicated as green in 2A). Now that we have the program, we can assess the selectivity and identity of mucin-domain glycoproteins that have been enriched (indicated by green + 'heat map' colors in 2C).

If this is not the case (more non-mucin proteins than predicted mucins in enriched portion) than I would consider that the enrichment possibly enriches for some other arrangement of glycosylated sites than the arrangements selected for by the domain candidacy algorithm.

StcE is a multidomain protein as demonstrated by its crystal structure (Yu *et al.*, *Structure*, 20(4), (2012)). The TXT motif is its cleavage motif within the protease's active site. However, as demonstrated by Nason *et al.*, *Nat Comm*, 12, 4070, (2021) the primary contributor to mucin binding is actually the C-terminal mucin binding domain. Thus, the TXT glycosite arrangement likely contributes little to mucin binding. We expect that any non-specific binding is from adherence to the POROS-AL beads and/or mucin/glycan binding proteins.

Additionally, though there are more non-mucin proteins enriched than predicted mucins, we have demonstrated that our enrichment technique is more selective for mucins than Jacalin enrichment (28% to 16% mucin proteins, respectively).

Specific questions about the mucin candidacy algorithm:

- While this phenomenon hasn't been systematically investigated, it is known that extracellular phosphorylation overlaps with O-GalNAc sites. This is most well-known on FGF23 (by FAM20C), but can also be seen on CD55/DAF, so removing GalNAc/phosphorylation site overlaps isn't 100% correct.

While we are aware that there are some instances of GalNAc/phospho-crosstalk, the number of false positive mucin-domain glycoproteins that we remove by making this distinction greatly outweighs losing a handful of true positives in the mucin-domain candidacy algorithm.

- The "12% rule" is defined as 12% of entire mucin domain, but the way that the mucin domain is derived isn't clarified (i.e., what decides the length of the mucin domain?). The length of the mucin domain will have an impact on whether this criterion is passed or not.

We apologize for the confusing language regarding the 12% rule. Here, the length of the mucin domain will not have an impact on whether the criterion is passed, because the 50 amino acid length is a sliding window through

the protein. The 12% rule is only analyzing how dense the glycosylation within these 50 amino acid windows actually is; the glycosylation cannot be more than 6 residues apart in order to be considered “dense enough” to be a mucin.

- Why add extra points for proteoglycan names (and mucin names)? Wouldn't this be better served by adding a whitelist for proteins that have the desired properties (e.g., CSPG) and can be classified as containing mucin-domains? Using names in this algorithm is fragile because protein names are subject to change, and as mentioned earlier, some proteins with mucin names aren't technically mucins.

This is a fair point. In considering this comment, we elected to remove the bonus points added for any protein name/descriptor. In doing so, we also re-evaluated the algorithm for proteins with known mucin domains to determine that the required O-glycosite count should be 9 instead of 10, and that the Mucin Score threshold should then be adjusted to 1.2 as the lower bound for what constitutes a mucin. We re-processed the dataset to account for this, and all data in the revised manuscript (including additional supplemental data added) reflects these changes.

General comments:

- There is a lot of other data available other than that published by Steentoft (SimpleCell) in 2013, and that data found in UniProt. Much more data has been published on the O-glycoproteome in the last 8 years, and includes data published by the Clausen lab, various serum O-glycoproteomic data sets, IsoTag data (Woo et al. 2017), and data from OpeRATOR enrichment (DOI: 10.15252/msb.20188486) which in principle also uses a mucinase to enrich (albeit for glycopeptides). It is likely that some of the novel mucin domain proteins have been already reported in more recent data sets.

While ideally we would use compiled O-glycosite/O-glycoprotein databases for this purpose, we had to use something available and tangible. We chose the SimpleCell dataset because it identified many O-glycopeptides from mucin-domain glycoproteins; some of these glycopeptides were found within our assigned mucin domains. Additionally, the sites were confirmed using electron-based fragmentation, unlike several other O-glycoproteomic datasets available (e.g. OpeRATOR enrichment). Further, the cell lines were not subjected to metabolic oligosaccharide engineering, which might affect the expressed surface proteins and/or glycans (e.g. IsoTag). Thus, we believe that the SimpleCell dataset was the best selection for our purposes.

- Shon et al. 2020, PNAS indicated that BT4244 could bind to K562 after sialic acid treatment. Did the authors consider testing the enrichment strategy using BT4244 on K562 after sialidase, or on COSMC knockout K562 (+ sialidase) to confirm that BT4244 can similarly enrich glycoproteins?

We thank the reviewer for this suggestion. To address this comment, we enriched mucin-domain glycoproteins using BT4244 mutant using sialidase treated HeLa cell lysate. As seen in Supplementary Figs 2 and 4B, sialidase pretreatment did aid in the enrichment of mucin-domain glycoproteins. The number of mucins went up from 6 to 13 (S2), and the selectivity from 10% to 20% (S4B). Unfortunately, it appears as if BT4244 is less robust at enrichment of mucins, especially when compared to StcE. However, we believe this serves as a proof-of-principle that other enzymes could potentially be used for the purpose of enrichment of mucins bearing truncated O-glycans.

- How does enrichment using an inactivated mucinase compare to enrichment using PNA or VVA? Does sialidase treatment affect the enrichment?

To answer reviewer 2, we performed an enrichment with Jacalin which is another lectin commonly used in enrichment for O-glycosylated proteins. The volcano plots for these enrichments are shown in Supplementary Figure 3. To be sure, Jacalin does enrich most of the mucin-domain glycoproteins, but as demonstrated by the large number of enriched non-mucin proteins, it is clear that Jacalin is less specific for mucin-domain glycoproteins. This point is further illustrated in Supplementary Figure 4. The Jacalin (+/-) pulldown resulted in the enrichment of 205 and 273 proteins, respectively. The percentage of mucin-domain glycoproteins within this subset is only 16-17%, meaning that 171 and 230 non-mucin proteins were found in the two samples.

Using the same HeLa lysate, StcE-conjugated beads enriched a total of 75 proteins, 28% of which were mucin-domain glycoproteins. Thus, StcE is 200% more selective for mucin-domains. Further, we detected only 54 non-mucin proteins in this enrichment, compared to the 230 in the Jacalin pulldown. This is an over 4-fold reduction in non-mucin proteins. Given that our ultimate goal is to identify glycopeptides from the mucin-domain glycoproteins, selectivity is extremely important. Non-mucin proteins, and their associated unmodified peptides, will outcompete the glycopeptides for ionization and detection.

This language and accompanying Supplementary Figures have been added to the manuscript.

- Figure 3A-D, Are there mucin-like proteins that are not enriched reproducibly? Labels for these proteins would be useful as investigations as to why they are not enriched could be interesting.

We appreciate this suggestion from the reviewer and have added **Supplementary Fig. 6** and **Data 5** to address this question. As with the enriched non-mucins, we generated an Upset Plot to determine which mucin-domain glycoproteins were not enriched reproducibly. Here, five proteins were consistently not enriched across all five cell lines and five across at least four. The grand majority of these proteins are intracellular cytoplasmic proteins that are likely to be overscored as mucins due to their phosphorylation sites. We have tried to account for these proteins by removing annotated phosphosites from the NetOGlyc glycosite assignments, though we recognize that the algorithm is still imperfect.

REVIEWERS' COMMENTS

Reviewer #1 (Remarks to the Author):

The authors addressed my questions and comments in the rebuttal. They included additional data to the manuscript and the presentation of the data has been improved. I only have a few minor additional comments and questions.

On the volcano plots of cell lysates (see Figures 3, SupplF 2B) yellow and orange dots seem to be more pronoucnly “depleted” in the enriched samples indicating that the mucin score threshold might be too permissive. On the other hand, in the ascites samples the vast majority of the predicted mucin-like proteins are enriched. What can be the reason for this “discrepancy”? Please comment.

As a corroboration of the selectivity of the mutant StcE towards mucin-like domains the authors performed a Jacalin enrichment with or w/o sialidase treatment and found that StcE is more selective. These results are summarized on SupplF 4. I find the the Fig caption “Jacalin exhibited poor selectivity for mucin-domain glycoproteins” a bit of an overstatement. Jacalin showed 16-17% selectivity compared to the average 25% selectivity of StcE. Is it really that big of a difference? Also, “BT4244 was 2x more selective”... right before results of Jacalin enrichment suggests that BT4244 is more selective under given experimental setup than Jacalin – based on the numbers, Jacalin is roughly just as selective as BT4244 on desialylated samples. Please reword the caption to better reflect the findings.

Furthermore, selectivity is defined as the percentage of target/off-target analytes captured by the method. I think the authors use this term incorrectly. For example, when they compare StcE and Jacalin enrichment, they argue that StcE is almost 2* as selective as Jacalin. I acknowledge that 28% is more than 16-17% but I think that the key factor here is not the difference in % (=selectivity) but the number of proteins present in the enriched samples, e.g. StcE is “better” as there are fewer proteins in the enriched samples therefore facilitating the identification of all proteins/peptides present including mucin-domain proteins. Similarly, it is said that “we detected only 54 non-mucin proteins in this enrichment, compared to the 230 in the Jacalin pulldown, representing a >4-fold reduction in non-mucin proteins. This selectivity is especially important”... , however the selectivity is still 28% as opposed to 16-17% with Jacalin.

The revised Figure 6 better captures the results of glycopeptide identifications. However, while it is evident that both the peptides and the glycans are quite different before and after enrichment, it would be interesting to see key players. I think the easiest way to capture it would be to add additional data in Suppl Datasets 11 and 12 including PSM# separately for the ascites and elute samples at the “glycan nodes” and “glycopeptides nodes” tabs. Also, I think it would be a nice

addition to briefly present/discuss the glycan structures derived from glycopeptide identifications. What are the major structures? Are these in good agreement with those expected/identified from similar samples? Are there published data on O-glycan studies in ascites? Furthermore, albeit out of the scope of the manuscript, the authors may comment on the ambiguity of these identifications. For example, N4H5A1F2 is among the top5 most abundant N-glycans, albeit these are probably misassignments of the same peptide with the more likely N4H5A2 structure due to wrong monoisotopic peak assignment of high m/z glycopeptide precursors. The same goes to N4H5F2 that is most likely a misinterpretation and the correct glycan is N4H5A1.

Referee 2 asked to present the mucin domain-related peptide identifications. The presented data currently do not allow the reader to see the predicted mucin domains of the proteins annotated as mucins. It would be nice to include this info as supplementary information. Please provide a list of protein sequences highlighting the predicted mucin domains (at least for all the identified mucin-domain proteins).

Remark on sample preparation: the strong interaction between StcE and target proteins is so strong that it calls for elution with SDS, therefore the in-gel digestion of the samples is the straightforward approach for proteolysis of the enriched samples. I do not argue that this is a valid approach but I suggest to try S-Trap digestion to see if it can streamline processing of higher number of samples in the future. I acknowledge that this would result in more complex peptide samples. Furthermore – why is the mutant StcE gives the most strongly stained spot in the SDS-PAGE separation (see SFig 1B)? The bait was covalently bound onto the POROS resin and excess protein was washed out according to the protocol in the Methods section.

On the „comparison” tab of Suppl. Dataset 10 lines there are 602 peptide-Oglycan combinations with the PSM# in elute and ascites samples. 220 such combinations (lines 384-602) were identified with 0 PSM in both samples. Why were these data included in the table? Maybe the summary page should be revisited by the authors.

A short description of the data published as „other supplemental materials” is listed on the front page of the pdf of the supplementary figures, however I think it would be much easier for the readers if the respective files themselves had a header line on the summary page, for example for Suppl. Dataset 10 on tab summary please include „O-glycopeptides identified from ovarian cancer patient ascites fluid” or something similar. Please correct.

Fig Legend of SF2: replace reductive amidation with reductive amination.

Reviewer #2 (Remarks to the Author):

I am happy the authors took my challenge and compared their method with the previous Jacalin method, highly improving the manuscript.

All other points raised has been addressed and corrected.

My only remaining point is that the legend of Figure 4 could be improved to better explain the different parts as done in the main text and in the review response.

Reviewer #3 (Remarks to the Author):

Thanks to the authors for addressing the concerns I had with the manuscript. I am satisfied with most of the responses and changes.

On my comments for figures 2A/C, I was suggesting a breakdown of the enrichment by mucin/non-mucin, which is well served by the new Supplementary figures 4 & 7.

However, a few points brought up by the changes could be addressed:

1) Can you make the y-axis on supplemental figure 4 the same scale so the bar heights are directly comparable between panel A and B? This way the enrichments (for HeLa) can be more easily directly compared.

2) I investigated a bit more into the predicted mucins that weren't enriched, and given that most of those hits are nucleocytoplasmic proteins, they all lack a signal peptide (and so should not pass through the secretory pathway). Could you add the presence of the signal peptide to your mucin candidacy algorithm? For these purposes, Signalp or retrieval of annotations from UniProt with the query string "annotation:(type:signal)" will work. This would reduce the number of false-positive hits from the algorithm.

Reviewer #1 (Remarks to the Author):

The authors addressed my questions and comments in the rebuttal. They included additional data to the manuscript and the presentation of the data has been improved. I only have a few minor additional comments and questions.

We again thank the reviewer for their time and effort spent on this manuscript.

On the volcano plots of cell lysates (see Figures 3, SupplF 2B) yellow and orange dots seem to be more pronoucnly “depleted” in the enriched samples indicating that the mucin score threshold might be too permissive. On the other hand, in the ascites samples the vast majority of the predicted mucin-like proteins are enriched. What can be the reason for this “discrepancy”? Please comment.

One potential reason for this is currently described in the text: “mucin domains in these proteins are not heavily glycosylated in [certain cell types]”. In other words, these proteins may have a high number of predicted O-glycosites, but they may not actually be glycosylated in a given cell type, meaning they would not be enriched by StcE^{E447D}. Another possibility is that biofluids have fewer proteins present that will interfere with mucin-domain glycoprotein binding. As a corollary, cell lysates have far more proteins that can (and will) non-specifically bind to the StcE-conjugated beads, blocking potential interactions with the mucin-domain glycoproteins. To address this additional reason, we have added the following text:

“The enrichment was likely more successful due to the presence of fewer interfering proteins found in biofluids.”

As a corroboration of the selectivity of the mutant StcE towards mucin-like domains the authors performed a Jacalin enrichment with or w/o sialidase treatment and found that StcE is more selective. These results are summarized on SupplF 4. I find the the Fig caption “Jacalin exhibited poor selectivity for mucin-domain glycoproteins” a bit of an overstatement. Jacalin showed 16-17% selectivity compared to the average 25% selectivity of StcE. Is it really that big of a difference? Also, “BT4244 was 2x more selective”... right before results of Jacalin enrichment suggests that BT4244 is more selective under given experimental setup than Jacalin – based on the numbers, Jacalin is roughly just as selective as BT4244 on desialylated samples. Please reword the caption to better reflect the findings.

We apologize for the misunderstanding; the sentence regarding BT4244 being “2x more selective” was only comparing the BT4244 enrichments before and after sialidase treatment. We have changed this sentence in the figure caption:

“Given BT4244’s inability to accommodate sialic acid, the enrichment was 2x more selective for mucins after sialidase treatment.”

Furthermore, selectivity is defined as the percentage of target/off-target analytes captured by the method. I think the authors use this term incorrectly. For example, when they compare StcE and Jacalin enrichment, they argue that StcE is almost 2* as selective as Jacalin. I acknowledge that 28% is more than 16-17% but I think that the key factor here is not the difference in % (=selectivity) but the number of proteins present in the enriched samples, e.g. StcE is “better” as there are fewer proteins in the enriched samples therefore facilitating the identification of all proteins/peptides present including mucin-domain proteins. Similarly, it is said that “we detected only 54 non-mucin proteins in this enrichment, compared to the 230 in the Jacalin pulldown, representing a >4-fold reduction in non-mucin proteins. This selectivity is especially important”... , however the selectivity is still 28% as opposed to 16-17% with Jacalin.

Selectivity refers to the extent to which the method can be used to determine particular analytes in mixtures or matrices without interferences from other components of similar behavior. Thus, we respectfully disagree with the reviewer that our definition of selectivity is incorrect. However, we have added a phrase that acknowledges the higher number of mucins in the Jacalin pulldown:

“While Jacalin did enrich more mucin-domain glycoproteins, selectivity is especially important when considering potential goals of characterizing mucin-domain O-glycopeptides.”

The revised Figure 6 better captures the results of glycopeptide identifications. However, while it is evident that both the peptides and the glycans are quite different before and after enrichment, it would be interesting to see key players. I think the easiest way to capture it would be to add additional data in Suppl Datasets 11 and 12 including PSM# separately for the ascites and elute samples at the “glycan nodes” and “glycopeptides nodes” tabs.

We thank the reviewer for this useful suggestion. We have modified Datasets S12 and S13 (formerly S11 and S12) to include PSM counts for ascites and elute conditions for both glycopeptides and glycans.

Also, I think it would be a nice addition to briefly present/discuss the glycan structures derived from glycopeptide identifications. What are the major structures? Are these in good agreement with those expected/identified from similar samples? Are there published data on O-glycan studies in ascites?

We agree that this would be interesting. Unfortunately, because we used HCD for fragmentation, we are unable to definitively determine glycan structures, especially because we cannot comment on whether glycan compositions are from a single glycan or the combination of multiple glycans. We modified the text to reference previous glycomics work (three new references) that can provide best guesses to structures that could match compositions:

“Even so, collision-based fragmentation can still provide O-glycopeptide identifications that include peptide sequence and the total glycan mass modification, though details about number of glycans or glycosite positions (and by extension, fine details about glycan structure) are usually inaccessible. Previous glycomics work suggests that some of these structures may include large, highly fucosylated and sialylated complex and hybrid N-glycans in addition to highly sialylated core-1 and-2 O-glycans with a smaller amount of sulfated core-2 O-glycan structures^{62–64}.”

Furthermore, albeit out of the scope of the manuscript, the authors may comment on the ambiguity of these identifications. For example, N4H5A1F2 is among the top5 most abundant N-glycans, albeit these are probably misassignments of the same peptide with the more likely N4H5A2 structure due to wrong monoisotopic peak assignment of high m/z glycopeptide precursors. The same goes to N4H5F2 that is most likely a misinterpretation and the correct glycan is N4H5A1.

We have added a sentence regarding this ambiguity:

“We note that there is some level of ambiguity in glycopeptide identifications, given that 2 fucose residues may be assigned as a single sialic acid and vice versa.”

Referee 2 asked to present the mucin domain-related peptide identifications. The presented data currently do not allow the reader to see the predicted mucin domains of the proteins annotated as mucins. It would be nice to include this info as supplementary information. Please provide a list of protein sequences highlighting the predicted mucin domains (at least for all the identified mucin-domain proteins).

Thank you for this suggestion. We have added a new dataset called Dataset S2 Mucin Domain Locations that shows the location and predicted O-glycosites that exist for each predicted mucin domain. This provides the reader to easy access examine where mucin domains occur and to understand where specific peptide sequences may fall relative to these mucin domains. The supplementary dataset numbering has been updated to account for this change, and the text has been updated to say:

“See **Supplementary Data 1** for the mucin candidate algorithm output of the entire human proteome and **Supplementary Data 2** for the location of where mucin domains and predicted O-glycosites occur.”

Remark on sample preparation: the strong interaction between StcE and target proteins is so strong that it calls

for elution with SDS, therefore the in-gel digestion of the samples is the straightforward approach for proteolysis of the enriched samples. I do not argue that this is a valid approach but I suggest to try S-Trap digestion to see if it can streamline processing of higher number of samples in the future. I acknowledge that this would result in more complex peptide samples.

We appreciate the reviewer's suggestion and are currently investigating alternative avenues for elution.

Furthermore – why is the mutant StcE gives the most strongly stained spot in the SDS-PAGE separation (see SFig 1B)? The bait was covalently bound onto the POROS resin and excess protein was washed out according to the protocol in the Methods section.

We agree that the leeching of StcE from the beads is curious, and again, this is an active area of research in our laboratories. We are testing alternative elution methods to maximize protein elution while minimizing StcE leeching.

On the „comparison” tab of Suppl. Dataset 10 lines there are 602 peptide-Oglycan combinations with the PSM# in elute and ascites samples. 220 such combinations (lines 384-602) were identified with 0 PSM in both samples. Why were these data included in the table? Maybe the summary page should be revisited by the authors.

We thank the reviewer for pointing this out. This was an error in transferring data. We reinspected all components of Figure 6 and associated data (newly numbered Datasets S10-S13) and saw that this error only affected Dataset S11 (formerly S10), which affected the calculations of the percent of O-glycopeptides that were from mucin domain glycoproteins, as discussed in the text and shown in Supplementary Figure 9. We fix the transfer error and reperformed calculations. The O-glycopeptide data now more closely matches the N-glycopeptide data, where the majority of identifications in the enriched samples are from mucin-domain glycoproteins, while the majority of identifications in the unenriched ascites fluid samples are not from mucin-domain glycoproteins. We have updated the text, Supplementary Figure S9, and Dataset S11 to reflect these changes.

“Similarly, approximately two thirds (~66%) of all O-glycopeptide identifications in the enriched samples belonged to mucin-domain glycoproteins, with only ~10% of O-glycopeptide identifications from unenriched ascites fluid samples deriving from mucin-domain glycoproteins (**Supplementary Fig. 9**).”

A short description of the data published as „other supplemental materials” is listed on the front page of the pdf

of the supplementary figures, however I think it would be much easier for the readers if the respective files themselves had a header line on the summary page, for example for Suppl. Dataset 10 on tab summary please include „O-glycopeptides identified from ovarian cancer patient ascites fluid” or something similar. Please correct.

We have now updated the legend in the supplemental tables to the following:

Supplementary Data 9: N-glycopeptide identifications from ascites fluid
Supplementary Data 10: O-glycopeptide identifications from ascites fluid

Fig Legend of SF2: replace reductive amidation with reductive amination.

We thank the reviewer for pointing out this error, we have made this correction.

Reviewer #2 (Remarks to the Author):

I am happy the authors took my challenge and compared their method with the previous Jacalin method, highly improving the manuscript.

All other points raised has been addressed and corrected.

We thank the reviewer for their kind comments and for their time spent reading the manuscript.

My only remaining point is that the legend of Figure 4 could be improved to better explain the different parts as done in the main text and in the review response.

We have now added the following text to the legend of Figure 4:

Figure 4. Analysis of mucin-domain glycoproteins from cell line enrichments. (A) Upset plot comparing enriched mucin lists from five cell lines. The total number of enriched mucin-domain glycoproteins from each cell line is shown on the bottom left (blue horizontal bars). If a group of mucin-domain glycoproteins was only seen in one cell line, only one gray dot is darkened; the number of proteins that are only seen in that cell line are shown in bar graph form above. Overlap between samples are shown by multiple darkened gray dots and a line connecting them. A total of seven mucin-domain glycoproteins were seen in all five cell lines; these proteins are shown above the Upset plot. The seven proteins found in all five samples are shown above the plot; putative mucin domain (orange), transmembrane domains (purple), and N-glycan sites (green) are annotated based on Uniprot assignments (transmembrane domain, N-glycan sites) and the mucin-domain candidacy algorithm (mucin domain).

Reviewer #3 (Remarks to the Author):

Thanks to the authors for addressing the concerns I had with the manuscript. I am satisfied with most of the responses and changes.

On my comments for figures 2A/C, I was suggesting a breakdown of the enrichment by mucin/non-mucin, which is well served by the new Supplementary figures 4 & 7.

We thank the reviewer for their kind comments and for their time spent reading the manuscript.

However, a few points brought up by the changes could be addressed:

1) Can you make the y-axis on supplemental figure 4 the same scale so the bar heights are directly comparable between panel A and B? This way the enrichments (for HeLa) can be more easily directly compared.

We have now included StcE in the bar graph of Supplemental Figure 4B for ease of comparison:

We also updated the figure legend:

“The StcE^{E447D} enrichment of HeLa lysate is shown for ease of comparison.”

2) I investigated a bit more into the predicted mucins that weren't enriched, and given that most of those hits are nucleocytoplasmic proteins, they all lack a signal peptide (and so should not pass through the secretory pathway). Could you add the presence of the signal peptide to your mucin candidacy algorithm? For these purposes, Signalp or retrieval of annotations from UniProt with the query string “annotation:(type:signal)” will work. This would reduce the number of false-positive hits from the algorithm.

We thank the reviewer for looking into this further and making this suggestion. Unfortunately, it is not a completely straightforward idea to implement. Upon looking into the signal peptide information available in Uniprot, not all transmembrane proteins have the signal peptide annotated for easy filtering. For example, dystrophin (P11532) and probable maltase-glucoamylase 2 (Q2M2H8) are two entries that are annotated as mucins by our algorithm and look by manual inspection to have mucin-like domains, but do not have signal peptide sequences in Uniprot. Thus, properly accounting for signal peptide information will require nuanced decision making to be properly incorporated into our workflow. We continue to look to improve the candidacy algorithm, so we will work to incorporate a version of this idea into further iterations outside the scope of this manuscript after we have thoroughly investigated how this information will affect outcomes.